# Mammalian orthoreovirus can exit cells in extracellular vesicles

**Sydni Caet Smith[1], Evan Krystofiak[2], Kristen M. Ogden**[1,3]*

1 Department of Pathology, Microbiology, and Immunology, Vanderbilt University Medical Center, Nashville, Tennessee, United States of America, 2 Department of Cell & Developmental Biology, Vanderbilt University, Nashville, Tennessee, United States of America, 3 Department of Pediatrics, Vanderbilt University Medical Center, Nashville, Tennessee, United States of America

* kristen.ogden@vumc.org

**Data Availability Statement:** All relevant data are within the manuscript and its Supporting Information files.

**Funding:** Research reported in this work was supported by the National Institutes of Health

## Abstract

Several egress pathways have been defined for many viruses. Among these pathways, extracellular vesicles (EVs) have been shown to function as vehicles of non-lytic viral egress. EVs are heterogenous populations of membrane-bound structures released from cells as a form of intercellular communication. EV-mediated viral egress may enable immune evasion and collective viral transport. Strains of nonenveloped mammalian orthoreovirus (reovirus) differ in cell lysis phenotypes, with T3D disrupting cell membranes more efficiently than T1L. However, mechanisms of reovirus egress and the influence of transport strategy on infection are only partially understood. To elucidate reovirus egress mechanisms, we infected murine fibroblasts (L cells) and non-polarized human colon epithelial (Caco-2) cells with T1L or T3D reovirus and enriched cell culture supernatants for large EVs, medium EVs, small EVs, and free reovirus. We found that both reovirus strains exit cells in association with large and medium EVs and as free virus particles, and that EV-enriched fractions are infectious. While reovirus visually associates with large and medium EVs, only medium EVs offer protection from antibody-mediated neutralization. EV-mediated protection from neutralization is virus strain- and cell type-specific, as medium EVs enriched from L cell supernatants protect T1L and T3D, while medium EVs enriched from Caco-2 cell supernatants largely fail to protect T3D and only protect T1L efficiently. Using genetically barcoded reovirus, we provide evidence that large and medium EVs can convey multiple particles to recipient cells. Finally, T1L or T3D infection increases the release of all EV sizes from L cells. Together, these findings suggest that in addition to exiting cells as free particles, reovirus promotes egress from distinct cell types in association with large and medium EVs during lytic or non-lytic infection, a mode of exit that can mediate multiparticle infection and, in some cases, protection from antibody neutralization.

## Author summary

The exit strategy that many viruses use to escape cells is unknown. Reovirus is a nonenveloped human virus and an ideal model system to understand virus exit strategies and their

(R01AI155646 to KMO; 1F31AI167541 to SCS), the National Center for Advancing Translational Sciences (CTSA Award no. UL1 TR002243), and by Dolly Parton Pediatric Infectious Diseases Research Funds. SCS was supported by the Chemical Biology of Infectious Diseases Training Program (NIH T32AI112541). Services at the Vanderbilt Cell Imaging Shared Resource performed through Vanderbilt University Medical Center's Digestive Disease Research Center were supported by National Institutes of Health grant P30DK058404 Core Scholarship. The Cell Imaging Shared Resource and EK were also supported by NIH CA68485, DK20593, DK58404, DK59637 and EY08126. The contents of this publication are solely the responsibility of the authors and do not necessarily represent the views of the National Institutes of Health or the National Center for Advancing Translational Sciences. The funders had no role in study design, data collection and analysis, decision to publish, or preparation of the manuscript.

**Competing interests:** The authors have declared that no competing interests exist.

influence on infection. We found that two different reovirus strains, one that disrupts cell membranes and one that leaves cells largely intact, increase the release of extracellular vesicles (EVs) from cells. Both reovirus strains are released from cells as free particles and in association with EVs, which are membrane-bound structures that function in cell-to-cell communication. Depending on cell type and virus type, EVs can act like an 'invisibility cloak' that shields reovirus from antibodies. EVs can also bundle and ferry reovirus particles between cells. Although we used cells to examine the effects of reovirus association with EVs, it is possible that in mammalian hosts, EVs may shield reovirus from immune defenses and promote more efficient transmission and infection through a 'strength-in-numbers' strategy. Future work building on these findings will test the biological significance of EV-enclosed reovirus and may inform delivery strategies for oncolytic reoviruses to tumor sites. Broadly, these findings enhance our understanding of virus egress strategies and infection principles that may apply to other viruses that travel in EVs.

## Introduction

Multiple viruses employ extracellular vesicles (EVs) as vehicles of non-lytic cellular egress [1,2]. EVs are membrane-bound structures released from cells, which remove cellular waste and function in intercellular communication by transporting molecules such as proteins, lipids, and nucleic acids [3–5]. Many subpopulations of EVs have been characterized and are generally differentiated based on their cell of origin, size, composition, and cellular biogenesis pathway [6]. Although EVs are highly heterogenous, and definitions change with expanding knowledge in the field, there are three broadly recognized EV categories: i) small exosomes (30–150 nm), ii) medium microvesicles (100–1000 nm), and iii) large apoptotic blebs (50–5000 nm) [6–10]. Several viruses, including BK polyomavirus, hepatitis A virus, and enterovirus 71 egress within exosomes [11–14]. Enteric viruses including bluetongue virus, poliovirus, and coxsackievirus egress within secretory autophagosomes, which are specialized medium EVs (300–900 nm) that form when double-membraned autophagosomes fuse with the plasma membrane to release single-membraned vesicles [15–21]. To date, the only virus released in microvesicles *in vitro* and *in vivo* is rotavirus, a member of the order *Reovirales* that causes acute gastroenteritis primarily in children [22]. Some viruses including rotavirus, Zika virus, and Epstein-Barr virus are capable of upregulating EV release, which may promote EV-mediated virus egress [23–25]. Despite recent discoveries, there remain many viruses whose egress mechanisms are poorly understood.

EV-mediated exit and transport are potentially advantageous strategies of cellular egress. For viruses such as BK polyomavirus, enterovirus 71, and hepatitis A virus, EV association can protect viral particles from antibody-mediated neutralization [11–13]. Phosphatidylserine, a phospholipid displayed on the surface of most EVs, serves as a potent down-regulator of anti-inflammatory immune responses and can function as a phagocytic uptake signal, increasing the likelihood that EV-associated viruses enter target cells [26,27]. EVs are additionally capable of enclosing multiple virus particles, which may enable collective virus infection of the same recipient cell [17,28]. This *en bloc* transport of viruses in EVs has been demonstrated both *in vitro* and *in vivo* [17,22,29].

Mammalian orthoreovirus (reovirus) is a nonenveloped virus with a segmented, double-stranded RNA genome. Mechanisms of reovirus egress are poorly understood. Like rotavirus and bluetongue virus, reovirus is a member of the order *Reovirales* which includes pathogens that cause disease in a wide range of human and animal hosts. Reovirus infects humans but is

rarely associated with disease [30,31]. Based on its capacity to lyse tumor cells, reovirus is currently in clinical trials as an oncolytic therapeutic [32]. Reovirus replication occurs in the cytoplasm [33]. Reovirus binds to specific cellular receptors and enters cells through the endocytic pathway. Binding and subsequent infection can be neutralized by strain-specific antibodies, some of which target and bind the reovirus attachment protein σ1 [34,35]. Following uptake, the virus is proteolytically uncoated and converted to an infectious subvirion particle (ISVP) via removal and cleavage of outer-capsid proteins and penetrates into the cytoplasm where the core synthesizes viral transcripts, which are translated [33]. Viral proteins accumulate in the cytoplasm and form replication factories that function as sites of particle assembly and maturation [33,36,37]. Mature particles are released from cells lytically or non-lytically, but it is currently unclear what mechanism controls these egress phenotypes [38]. Two strains of reovirus, type 1 Lang (T1L) and type 3 Dearing (T3D), differ in their pathogenesis, tropism, and capacity to induce apoptosis, with T3D inducing apoptosis more efficiently than T1L [39–43]. There appears to be little connection between apoptosis induction and progeny virus yield [40,44,45].

Reovirus infection initiates lysis in some types of cells, including HeLa cells and Madin-Darby canine kidney cells [45,46]; however, in human brain microvascular endothelial cells (HBMECs) and in primary human airway epithelia, reovirus exits cells in the absence of lysis [47–49]. HBMECs and primary human airway epithelia may more closely model the cells reovirus infects in animals than the immortal HeLa cell line. In HBMECs, lysosomally-derived membranous structures termed "sorting organelles" appear to gather mature reovirus particles from reovirus replication factories [48]. Groups of reovirus particles are then shuttled to the plasma membrane in smaller "membranous carriers," which fuse with the plasma membrane and release free reovirus particles non-lytically. While a non-lytic egress mechanism in one cell type has been characterized, non-lytic egress pathways in other cell types remain unknown.

In this study, we sought to determine how reovirus is released from infected cells and how reovirus egress strategies impact the downstream infection of recipient cells. Using two strains of reovirus that differ in membrane disruption efficiency, T1L and T3D, and a virus that we engineered to contain genetic barcodes, we show that reovirus infection enhances the release of EVs, and that reovirus can egress from cells in association with EVs. In some cases, these EVs shield reovirus particles from antibody-mediated neutralization and transmit multiple reovirus particles between cells. This work reveals a potential mechanism by which reovirus may escape immune system defenses and overcome cellular thresholds to infection, enhancing the likelihood of productive infection. These findings, which enhance our understanding of the influence of egress strategy on virus infection, may apply broadly to viruses that are released in EVs and travel in association with EVs. Further insights into the mechanisms and impacts of EV-mediated viral egress may help inform improved viral vaccination strategies and delivery of viral vectors, including oncolytic reovirus therapeutics.

## Results

### Reovirus protein co-fractionates with EV-enriched fractions released from cells irrespective of plasma membrane integrity phenotype

We evaluated the effects of T1L and T3D replication on plasma membrane integrity in murine L929 fibroblasts (L cells), which are susceptible to reovirus infection and produce high viral yields [50]. To assess reovirus replication efficiency, we adsorbed L cells with T1L or T3D reovirus and harvested infected cell culture supernatant every 24 h for a total of 96 h. We quantified total virus titer, including both virus replicating inside of cells and virus released from

cells, at each time point using a fluorescent focus assay (FFA). Although the inoculum contained identical infectious units, cell binding appeared to vary between the two strains, as T1L titer immediately after adsorption was significantly lower than that of T3D (**Fig 1A**). However, replication for both viruses was efficient and reached similar peak titers by 48 h post infection (p.i.). After adsorbing L cells with T1L or T3D reovirus or medium alone (mock), we evaluated plasma membrane disruption using trypan blue staining every 24 h for 96 h. Compared to T1L-infected and mock-infected cells, significantly more T3D-infected cells were trypan blue positive, with plasma membranes of nearly all cells disrupted by 96 h p.i. (**Fig 1B**). In contrast to T3D infection, T1L infection yielded low levels of trypan blue-positive cells comparable to mock infection at most time points, indicating minimal plasma membrane disruption. We also evaluated plasma membrane damage using a lactase dehydrogenase (LDH) assay. As observed using the trypan assay, fluorescence-based quantitation of LDH release indicated that T3D induces significantly more cellular cytotoxicity than T1L, which failed to induce more damage than medium alone (**S1 Fig**). Thus, although T1L and T3D both replicate efficiently in L cells, these strains exhibit significant differences in their capacity to disrupt cell membranes. Therefore, we suspected that these viruses may employ different egress strategies.

To evaluate reovirus association with fractions enriched for EVs, we adsorbed L cells with T1L or T3D and used sequential differential centrifugation to fractionate EV populations. The centrifugation conditions chosen enrich for certain sizes of EVs; $2,000 \times g$ enriches for large EVs, and $10,000 \times g$ enriches for medium EVs (**Fig 1C**) [6]. Centrifugation at $100,000 \times g$ is anticipated to pellet a mixed population of small EVs and free reovirus particles [6,50]. These fractions are not "pure" populations of any one type of EV; rather, they represent an enrichment based on size. However, we anticipate that apoptotic blebs would primarily be enriched in the large EV fraction, microvesicles in the medium EV fraction, and exosomes in the small EV fraction [6]. Due to their size and density, free reovirus particles are not anticipated to pellet at $2,000 \times g$ or $10,000 \times g$ unless they are directly associated with larger structures [50]. To determine whether reovirus protein associates with each EV-enriched fraction, we harvested supernatant from infected L cells every 24 h for 96 h and enriched for large EV, medium EV, and small EV/free virus fractions. We resolved equal volumes of each sample by SDS-PAGE and immunoblotting and quantified the reovirus λ3 structural protein signal associated with each fraction. We found that reovirus structural protein associated with fractions enriched for each EV size, and association increased with infection time (**Fig 1D–1G**). By 96 h p.i., we detected T1L protein in approximately equivalent proportion in association with large EV, medium EV, and small EV/free virus fractions (**Fig 1E**). Likewise, at 96 h p.i., we detected T3D protein associating approximately equivalently with medium EVs and with the small EV/free virus fraction, though T3D protein association with the large EV fraction was comparatively lower (**Fig 1G**). Thus, although some strain-specific protein association differences exist between the reovirus strains, we detect association of reovirus structural protein with supernatant fractions enriched for large EVs, medium EVs, and small EVs/free virus.

To ascertain whether reovirus nonstructural protein associates with each EV-enriched fraction, we harvested supernatant from infected L cells at 72 h p.i. and enriched for large EV, medium EV, and small EV/free virus fractions. We chose the 72 h p.i timepoint for this analysis because there are high amounts of detectable reovirus-EV association without the near complete plasma membrane disruption induced by T3D at 96 h p.i. (**Figs 1B, 1E, 1G** and **S1**). We resolved equal volumes of each sample by SDS-PAGE and immunoblotting and quantified the reovirus σNS nonstructural protein signal associated with each EV fraction (**S2A Fig**). We observed that T1L σNS protein (**S2B Fig**) associates only with the small EV/free virus fraction, whereas T3D σNS protein (**S2C Fig**) associates with both the medium and large EV fractions.

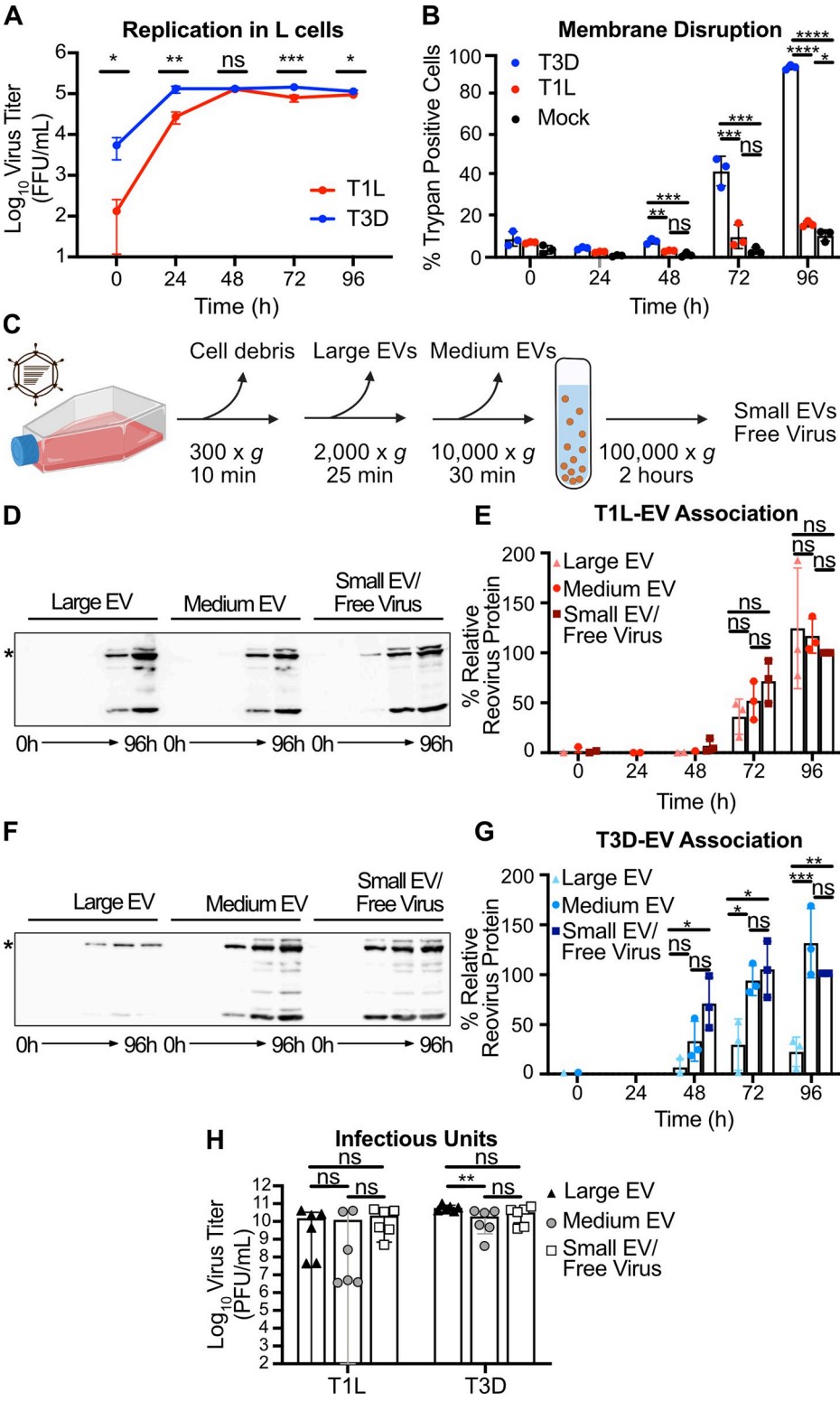

**Fig 1. Reovirus protein co-fractionates with EV-enriched fractions released from cells regardless of plasma membrane integrity phenotype.** L cells were adsorbed with three individual clones of T1L or T3D reovirus at an MOI of 1 PFU/cell. (A) Every 24 h, cell lysates were collected, and virus in the supernatant was quantified by FFA. Error bars indicate SD. $n = 3$. *, $P < 0.05$; **, $P < 0.01$; ***, $P < 0.001$ by two-sample unpaired T test. (B) Cell membrane disruption was quantified for T1L-, T3D-, and mock-infected cells every 24 h for 96 h using trypan blue staining. Error

bars indicate SD. $n$ = 3. *, P < 0.05; **, P < 0.01; ***, P < 0.001; ****, P < 0.0001 by one-way ANOVA with Tukey's multiple comparisons. (C) Schematic showing the EV fraction enrichment protocol described in the text. Created using Biorender.com. (D-G) Infected-cell supernatants were collected every 24 h for 96 h. Mock-infected supernatant was collected at 96 h, but reovirus protein was not detected. Reovirus protein association with large EV, medium EV, and small EV/free virus fractions was quantified following SDS-PAGE and immunoblotting. Representative immunoblots probed using reovirus antiserum for T1L (D) and T3D (F) and graphs showing results quantified from three independent immunoblots for T1L (E) and T3D (G) are shown. Asterisk denotes the reovirus λ3 protein band used for quantitation. Error bars indicate SD. $n$ = 3. *, P < 0.05; ***, P < 0.001 by one-way ANOVA with Tukey's multiple comparisons prior to normalization. Protein signal was normalized as a percentage of maximum by dividing each adjusted volume value by the highest measured value within the blot. (H) Infected-cell supernatants were harvested at 72 h, and viral infectious units associated with each EV fraction were quantified by plaque assay. Error bars indicate SD. $n$ = 3. **, P < 0.01 by two-way ANOVA with Tukey's multiple comparisons.

To determine whether the reovirus protein associated with EV fractions represented infectious reovirus, we used a plaque assay to determine the titers of T1L and T3D associated with large EV, medium EV, and small EV/free virus fractions. At 72 h p.i., infectious reovirus was detected in all EV fractions (Fig 1H). For T3D, we detected high infectious virus titers associated with fractions enriched for large EVs, medium EVs, and small EVs/free virus, with the most consistently high titers associated with large EVs (Fig 1G–1H). For T1L, infectious virus titers associated with fractions enriched for large EVs and medium EVs were variable and sometimes lower than those present in the small EV/free-virus fraction, though titers were generally high, and differences were not statistically significant (Fig 1H). Combined, these data suggest that, irrespective of the capacity to induce cell membrane disruption during infection, infectious T1L and T3D reovirus are released from cells in association with cell-derived structures harvested under conditions that enrich for EVs.

## Extracellular reovirus visually associates with large and medium EVs

To image the cell-derived structures with which reovirus associates, we adsorbed L cells with T1L or T3D reovirus. We harvested cell supernatants at 72 h p.i., enriched for large EVs or medium EVs by sequential centrifugation, and imaged each fraction using negative-stain transmission electron microscopy (EM). Large EV and medium EV fractions were generally enriched for the target EV size of interest but were not homogenous (Fig 2). Large EVs purified from reovirus-infected cell supernatants contained enveloped structures hundreds to more than a thousand nanometers in diameter with membranes that often appeared thin and non-uniform, potentially due to a loss of the contents within the large EVs (Fig 2A–2B). In some cases, the EV structures in this fraction were smaller, appeared to have thicker membranes, and formed aggregates. We observed reovirus particles measuring about 80 nm in diameter adhered to, or in some cases possibly enclosed within, these structures. When we visualized medium EVs purified from supernatants of reovirus-infected cells (Fig 2C–2D), we observed vesicles measuring ~ 600 nm in diameter. These EVs and tended to have rounder, more uniform shapes with well-defined membranes. We observed reovirus particles associating with medium EVs, though it was often unclear whether particles were on the interior or exterior of the EVs. We observed single viral particles, pairs of particles, and multiparticle clusters (Fig 2A–2D). Though our methodology was designed to limit this EV disruption, we cannot exclude the possibility that some EVs were disrupted during sample preparation for EM imaging and subsequently released virus particles. Overall, these findings suggest that centrifugation enriches for large and medium EVs, although these fractions do appear to contain at least partially heterogenous EV populations. Furthermore, T1L and T3D reovirus both associate with large and medium EVs; however, it is unclear whether the reovirus particles are bound on the exterior of the EVs or packaged internally.

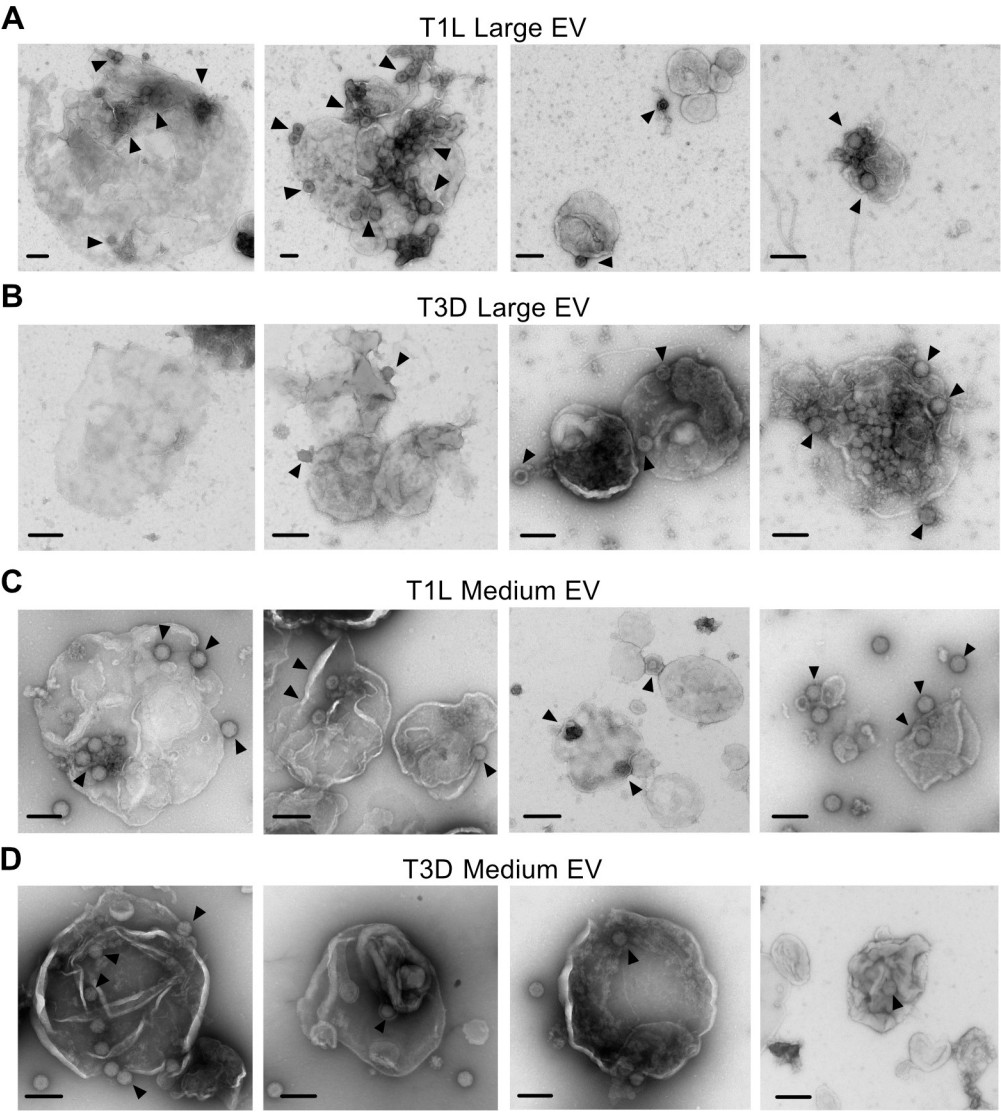

**Fig 2. Extracellular reovirus visually associate with large and medium EVs.** L cells were adsorbed with reovirus strain T1L or T3D at an MOI of 1 PFU/cell for 72 h. Cell supernatants were collected and sequentially centrifuged to enrich for large EVs (A-B) or medium EVs (C-D), which were visualized using negative-stain EM. Arrowheads indicate viral particles. Scale bars = 200 nm.

## Extracellular reovirus fails to associate with small EVs

We anticipated that the final step in the sequential centrifugation protocol enriched for a mixed population of small EVs and free reovirus particles. To determine whether small EVs could be separated from free reovirus particles, we infected L cells with T1L or T3D for 72 h, harvested the supernatant, and concentrated large EV-depleted and medium EV-depleted supernatant on an iodixanol cushion (**Fig 3A**). We applied the resulting small EV/free virus pellet to an iodixanol gradient and centrifuged overnight. We collected 12 × 1 ml fractions, with fraction 1 representing the top of the gradient and fraction 12 representing the bottom of the gradient. We resolved collected fractions and immunoblotted for reovirus proteins and a protein marker of small EVs, CD81 (**Fig 3B–3E**) [6,51]. Gradient-separated T1L-infected cell supernatants yielded a strong CD81-positive small EV signal in fractions 7–9, which was

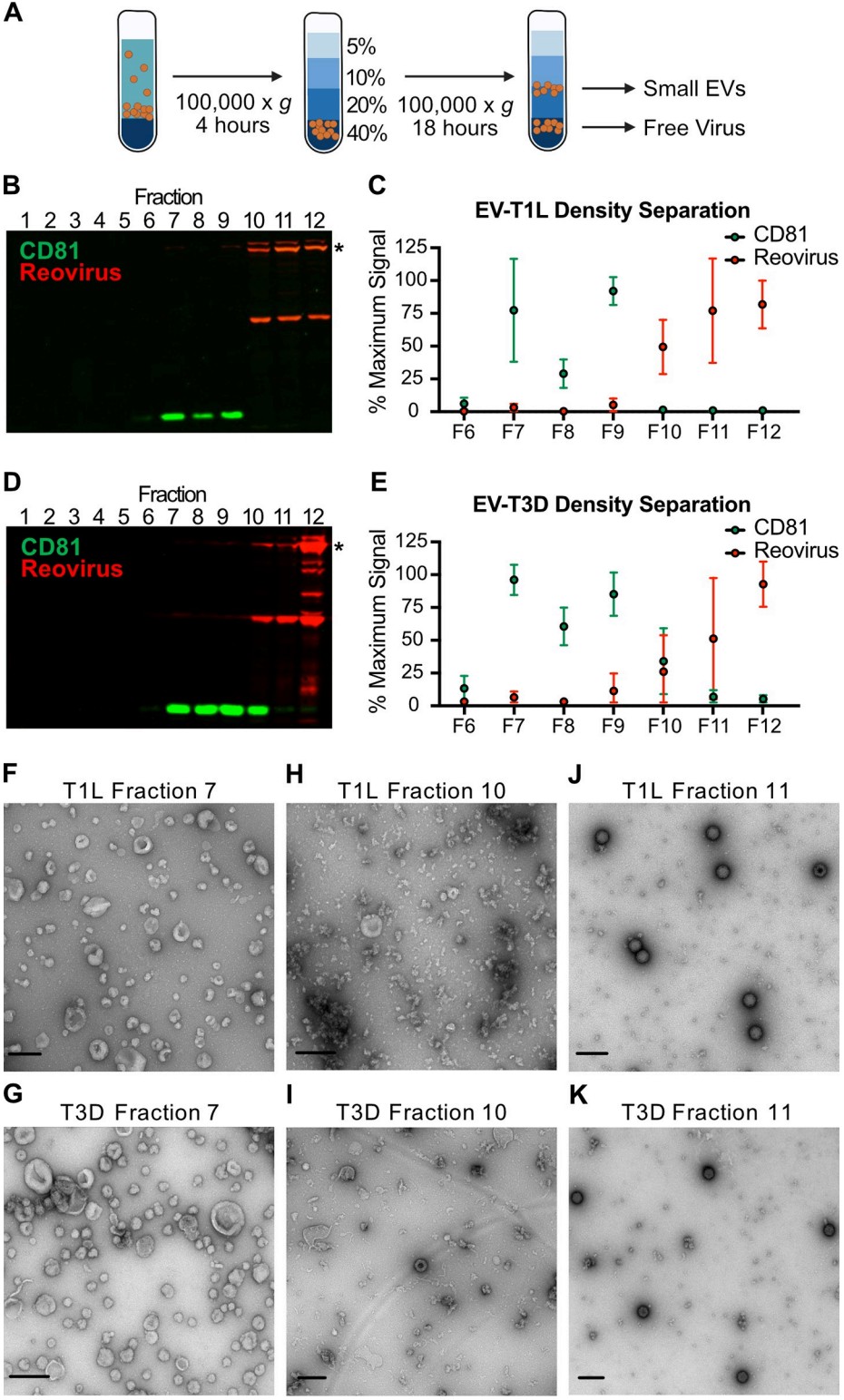

**Fig 3. Extracellular reovirus particles fail to associate with small EVs.** L cells were adsorbed with three individual clones of reovirus strains T1L or T3D at an MOI of 1 PFU/cell for 72 h. Cell debris, large EVs, and medium EVs were cleared from infected-cell supernatants, as in **Fig 1C**. (A) The resulting supernatant was centrifuged on a 60% iodixanol cushion to concentrate small EVs and free virus particles. The pellet was loaded onto a 5–40% iodixanol gradient. Twelve 1-ml fractions were collected and analyzed. Created using Biorender.com. (B-E) T1L-infected (B-C)

or T3D-infected (D-E) iodixanol gradient fractions were resolved using SDS-PAGE and immunoblotting to detect CD81 (green) and reovirus proteins (red). (B, D). Relative CD81 and reovirus protein signals in fractions 6–12 were quantified in three independent experiments. Asterisk denotes the reovirus λ3 protein band used for quantitation. (C, E). Error bars indicate SD. (F-K) Contents of fractions 7 (F, G), 10 (H, I), and 11 (J, K) were imaged using negative-stain EM. The reovirus strain used to infect the cells from which gradient-separated supernatant fractions were collected is indicated. Scale bars = 200 nm.

distinct from the reovirus protein signal detected in fractions 10–12 (**Fig 3B–3C**). Gradient-separated T3D-infected cell supernatants exhibited a similar phenotype; we detected CD81 in fractions 7–10 and reovirus proteins in fractions 9–12, with peak signals of each in distinct fractions (**Fig 3D–3E**). For T1L and T3D, fraction 7 contained small EVs, which resembled exosomes based on their small size and cup-shaped morphology [52,53]; we did not detect reovirus particles in this fraction (**Fig 3F–3G**). Fraction 10 contained mainly protein aggregates, with some small EVs scattered sparsely throughout (**Fig 3H–3I**). We did not detect any T1L particles in this fraction, and although we did detect T3D particles, we did not observe physical association of the reovirus particles with the small EVs. Fraction 11 contained free T1L and T3D virus particles, with no small EVs (**Fig 3J–3K**). These findings suggest that a subset of reovirus progeny egresses from L cells as free virus particles that fail to associate with any EV population.

## Nonspecific reovirus adhesion to large and medium EVs is inefficient

Next, to determine whether reovirus association with large or medium EVs is primarily mediated through nonspecific external adhesion, we evaluated the capacity for reovirus to associate with EV fractions harvested from mock-infected cells. We collected large EV and medium EV fractions from T1L-infected, T3D-infected, and mock-infected L cells at 72 h p.i. We incubated free T3D and T1L virus particles with equivalent volumes of large EVs and medium EVs harvested from mock-infected cells, or with vehicle buffer, and re-pelleted each fraction at the respective centrifugation speed. Using Coommassie blue staining, we determined that thrice the number of uninfected cells was required to yield approximately equivalent protein amounts in each uninfected EV-enriched fraction compared to reovirus-infected EV-enriched fractions (**S3A–S3B Fig**). Using SDS-PAGE and immunoblotting for reovirus protein, we found that the amount of protein from free T1L virus and free T3D virus that spontaneously associates with uninfected large and medium EVs was noticeably lower than the amount of reovirus that associates with EVs during infection, though we cannot directly compare input virus under the two conditions (**S3C and S3E Fig**). Free reovirus particles failed to pellet at $2,000 \times g$ or $10,000 \times g$ in the absence of EVs (**S3C–S3F Fig**) [50]. Furthermore, we observed that the spontaneous association of T1L virus with large and medium EVs from mock-infected cells was significantly lower than the free T1L virus input (**S3D Fig**). Interestingly, although the spontaneous association of T3D virus with large and medium EVs from mock-infected cells was also significantly lower than the free T3D virus input, the percent of input associated with EVs we detected was higher, suggesting more efficient adhesion of T3D than T1L to vesicles (**S3F Fig**). Though strain-specific differences in the efficiency of association exist, nonspecific virus adhesion to large and medium EVs is unlikely to fully explain the association of these particles with the EVs during egress.

## EV-mediated reovirus egress is consistent with microvesicle biogenesis

To visualize T1L and T3D egress from cells, we adsorbed L cells with T1L or T3D reovirus, and imaged using thin-section transmission electron microscopy (TEM) (**S4 Fig**). In rare

occurrences, we detected single T1L or T3D particles associating with plasma membrane structures that appeared to be budding outwards (**S4A–S4B Fig, left two images**). We observed T1L or T3D virus particles present as single particles, pairs of particles, or clusters of multiple particles present within EV-like structures that were distinct but proximal to the plasma membrane (**S4A–S4B Fig, right two images**). These EV-like structures ranged in diameter from ~ 400 nm to ~ 600 nm, similar to the size of microvesicles. Thus, the phenotype of at least one reovirus egress pathway appears consistent with the biogenesis of microvesicles, which bud outward from the plasma membrane prior to pinching off and release [3,7,10].

## Medium-sized EVs protect reovirus from neutralization and proteolysis

To determine whether association with EVs can shield reovirus from antibody-mediated neutralization, we employed a plaque reduction neutralization assay. We enriched large EV, medium EV, and small EV/free virus fractions, as well as iodixanol gradient-separated free virus, from the supernatants of L cells infected with T1L or T3D for 72 h (**Fig 4A**). We treated each fraction with reovirus strain-specific neutralizing antiserum or with medium alone and determined titers by plaque assay. We hypothesized that if reovirus is present as free particles or as particles adhered to the EV exterior, then the virus would be sensitive to neutralization, and the treated sample titer would be reduced relative to the untreated sample titer. However, if reovirus particles are enclosed within EVs, then the virus would be protected from neutralization, and the treated sample titer would be comparable to the untreated sample titer. We found that when reovirus associated with large EV or small EV/free virus fractions, both T1L and T3D were neutralized to similar levels as free reovirus, with titers reduced on average by 100-fold (**Fig 4B–4E**). In contrast, when associated with the medium EV fraction, T1L and T3D titers were unaffected, demonstrating robust protection from neutralization. These findings suggest that T1L and T3D particles released from L cells are specifically packaged inside medium EVs, but not inside large EVs.

In a complimentary approach, we used the protease chymotrypsin, which can covert reovirus virions to ISVPs *in vitro*, to assess the protective capacity of EVs for virus particles. Following chymotrypsin treatment, virions lose outer-capsid protein σ3, and µ1C is cleaved to fragments including δ [33]. Using a concentration of chymotrypsin at which we observed full conversion of virions to ISVPs in the small EV/free virus fraction, we treated large, medium, and small EVs enriched from infected L cells with 20 µg/mL of chymotrypsin. Then, we resolved viral structural proteins by SDS-PAGE and immunoblotting and quantified the σ3 or µ1C protein signal in chymotrypsin-treated samples relative to mock-treated samples (**S5A and S5C Fig**). For both T1L and T3D, the µ1C protein signal retained post chymotrypsin treatment compared to the mock-treated signal was significantly higher for reovirus associated with medium EVs than small EVs (**S5B and S5D Fig**). For T1L, this signal is also significantly higher than that associated with large EVs. Although the σ3 protein signal retained post chymotrypsin treatment was not statistically significant in the medium EV fraction compared with the other fractions, likely due to high variability, it was the only EV fraction in which σ3 signal was detected. Thus, it appears that medium EVs are capable of at least partially protecting released reovirus from two components of the extracellular environment, neutralizing antibodies and proteases.

## Large and medium EV-associated reovirus can mediate multiparticle infection

To determine whether EVs can transport an infectious unit consisting of multiple reovirus particles, we used high-resolution melt (HRM) analysis to detect genotypes of individual viral

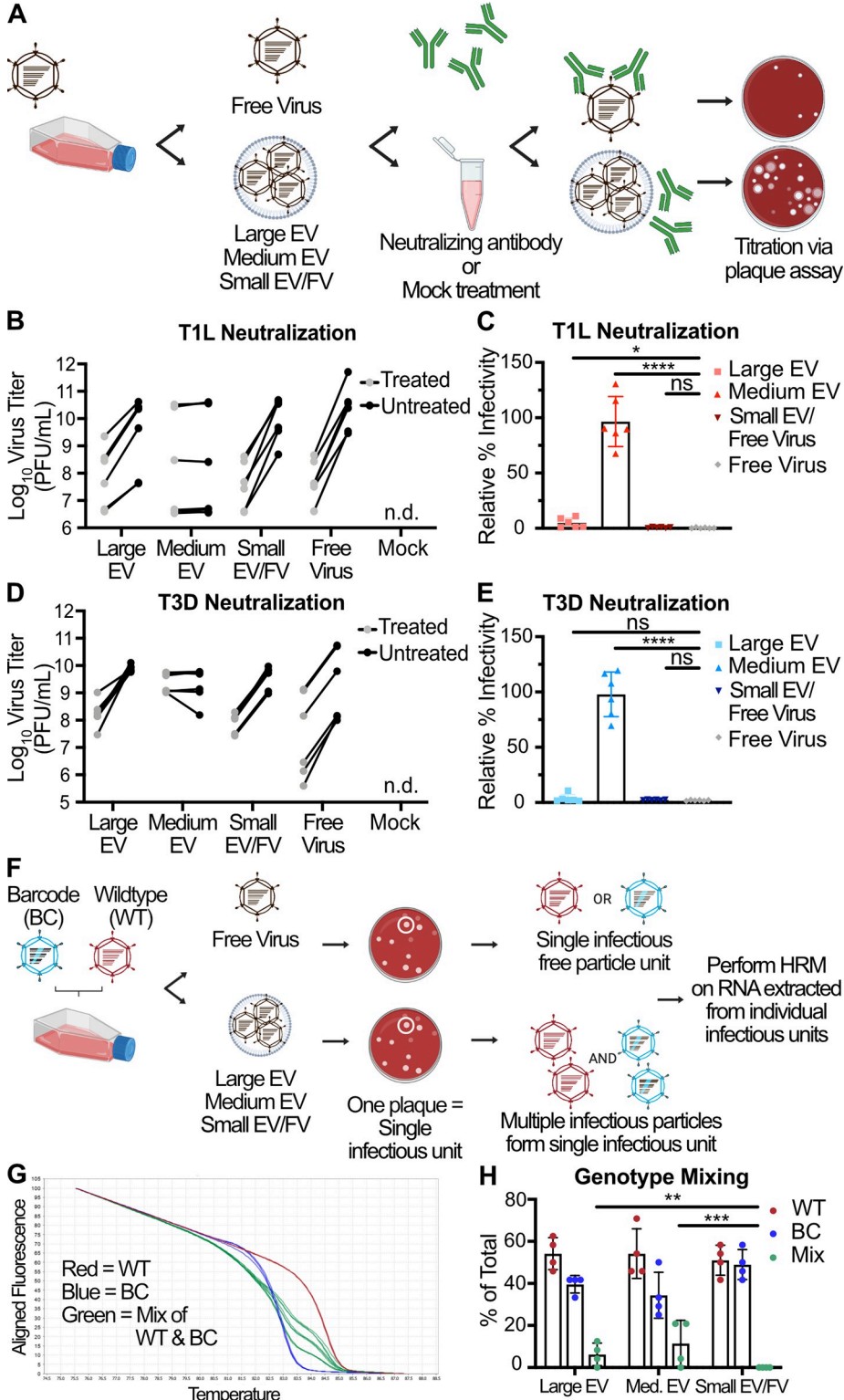

**Fig 4. Multiple reovirus particles can be transported by protective, medium-sized EVs and non-protective large EVs.** (A) L cells were adsorbed with three individual clones of reovirus strains T1L or T3D at an MOI of 1 PFU/cell. After 72 h, EV-associated and free reovirus particles were enriched using sequential centrifugation and iodixanol gradient separation, as previously described, then incubated with σ1-specific reovirus antiserum (treated) or with diluent (untreated). Infectious units were quantified by plaque assay. Created using Biorender.com. (B-E) Plaque titer

(B, D) and percent relative infectivity, quantified by dividing the treated infectious units by mock-treated infectious units and multiplying by 100 (C, E), for each sample are shown. Error bars indicate SD. "n.d." = not detected. $n$ = two titers per sample in each of three independent experiments. *, P < 0.05; ****, P < 0.0001 by two-sample unpaired T test. (F) L cells were coinfected with independent inocula of WT or BC T3D reovirus at an MOI of 10 PFU/cell. At 24 h p.i., large EV, medium EV, and small EV/free virus fractions were harvested from the supernatants using sequential centrifugation and subsequently used to inoculate a plaque assay. Plaques representing individual infectious units, which might be EV-associated bundles or free virus particles, were picked and amplified. Viral RNA was genotyped using HRM. Created using Biorender.com. (G) Normalized melt curves for control RNA from WT (red), BC (blue), and 2:1, 1:1, and 1:2 mixtures of WT and BC (green) RNA are shown. (H) Genotype quantitation for the large EV, medium EV, and small EV/free virus fractions as a percentage of total plaques analyzed. Error bars indicate SD. $n$ = 24 plaques represented by each data point in four independent experiments. **, P < 0.01; ***, P < 0.001 by Pearson's chi-squared analysis with pairwise comparisons.

particles [54]. We co-infected L cells with wild-type (WT) and genetically barcoded (BC) T3D reoviruses, collected cell culture supernatants at 24 h p.i., and enriched for large EV, medium EV, and small EV/free virus fractions (**Fig 4F**). We adsorbed fresh L cell monolayers with serially diluted intact EV fractions or free reovirus particles and isolated individual plaques, which we define here as single infectious units. In the case of a free virus particle, a single infectious unit is likely to be an independent WT particle or an independent BC particle. If an EV bundles multiple particles together, then a single infectious unit could contain multiple WT, multiple BC, or multiple WT and BC particles. We genotyped individual viral plaque infectious units using HRM analysis, which distinguishes WT and BC RNA based on differences in melt temperature conferred by genetic polymorphisms in the barcode (**Fig 4G**). A multiparticle population containing both WT and BC viruses is anticipated to yield an intermediate melt curve. A multiparticle population containing only WT or only BC viruses will yield a melt curve indistinguishable from that generated by infection by a single virus particle (**S6 Fig**). In each of four independent experiments, we examined 24 plaques from each fraction, for a total of 96 plaques representing each of the three fractions. While we detected no genotype mixtures in plaques formed from virus in the small EV/free virus fraction, a significant portion of reovirus plaques in the large EV-enriched fraction (~ 6%) contained multiple genomes (**Fig 4H**). In the medium EV-enriched fraction, we detected statistically significant and slightly higher levels of reovirus plaques containing multiple genomes (~ 11%). Contrastingly, multiple genomes were never detected in plaques formed by small EVs or free virus. Thus, our data suggest that a significant portion of large EVs and medium EVs, but not small EVs, can ferry reovirus between cells as infectious multiparticle bundles.

## EV-mediated reovirus egress occurs in multiple cell types, but neutralization protection may be cell type- and virus strain-dependent

To investigate whether EV-mediated reovirus egress occurs in cell types other than L cell murine fibroblasts, we used non-polarized Caco-2 human colon epithelial cells, which more closely resemble the intestinal epithelial cells reovirus infects in mammals [33]. We compared titers of T1L and T3D following infection of Caco-2 cells to those following infection of L cells during a time course of infection (**Fig 5A–5B**). Although T1L replicated significantly less efficiently in Caco-2 cells at a multiplicity of infection (MOI) of 5 PFU/cell than in L cells at an MOI of 1 PFU/cell, T1L titer increased throughout the time course by more than 500-fold (**Fig 5A**); T3D replication in Caco-2 cells and L cells was comparable (**Fig 5B**). Plasma membrane damage and disruption phenotypes for both viruses were similar, with T3D inducing significantly higher levels of membrane disruption compared to T1L (**Figs 1B, 5C and S7**). While T1L protein association was equivalent among all fractions in L cells, T1L protein association with Caco-2-derived medium EV fractions was significantly less relative to small EV/free virus

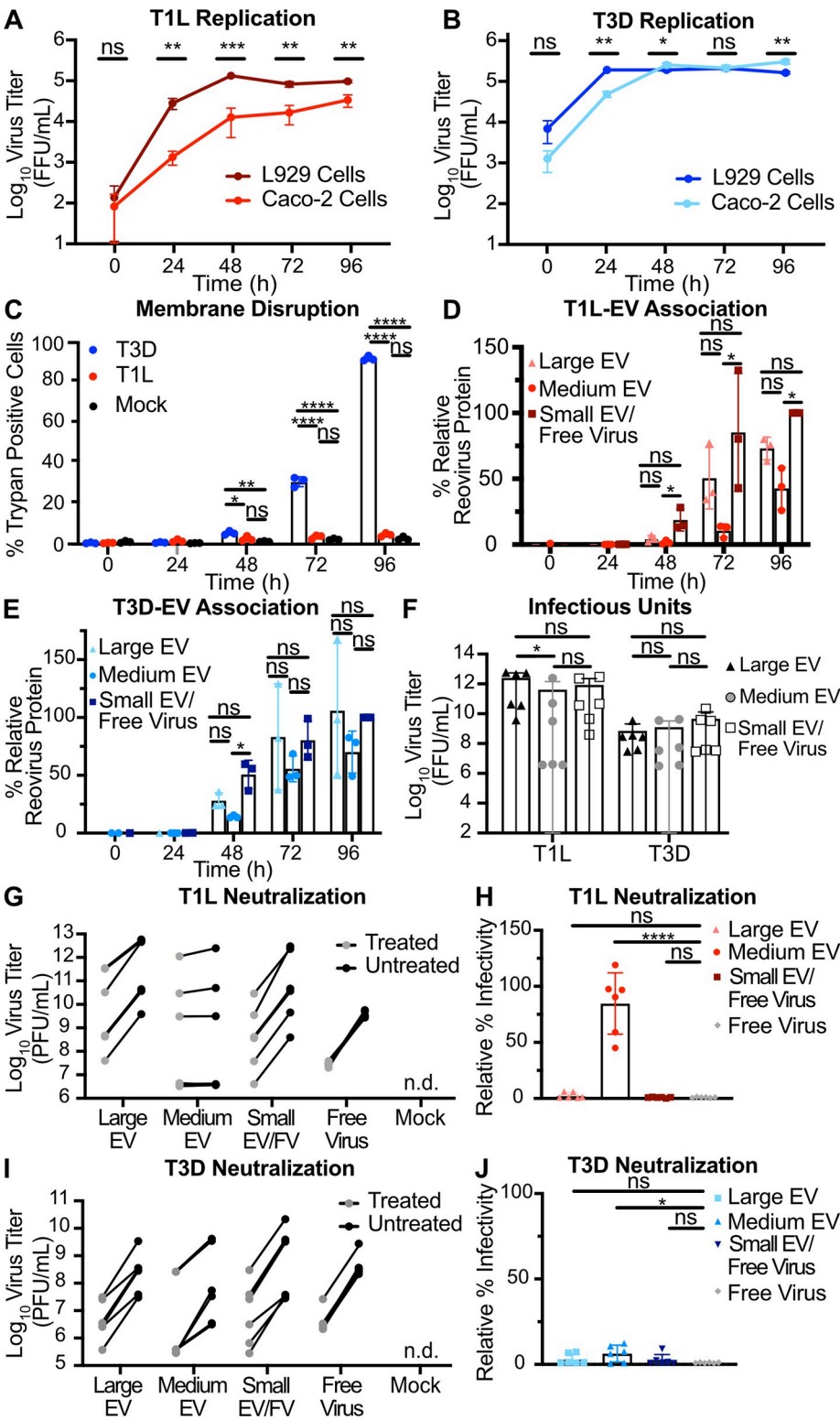

**Fig 5. EV-mediated reovirus egress occurs in multiple cell types, but protection from neutralization may be strain dependent.** (A-B) Caco-2 cells were infected with three individual clones of T1L (A) or T3D (B) at an MOI of 5 PFU/cell (Caco-2), as quantified in L cells. At the indicated timepoints, cell lysate was collected, and virus titers were determined by FFA. Results for T1L or T3D infection of L cells at an MOI of 1 PFU/cell are duplicated from **Fig 1A** for comparison. Error bars indicate SD. $n = 3$. *, $P < 0.05$; **, $P < 0.01$; ***, $P < 0.001$ by two-sample unpaired T test. (C)

Caco-2 cells were adsorbed with three individual clones each of reovirus strains T1L or T3D at an MOI of 5 PFU/cell. Cell membrane disruption was quantified for T1L-, T3D-, and mock-infected cells every 24 h for 96 h using trypan blue staining. Error bars indicate SD. $n = 3$. *, $P < 0.05$; **, $P < 0.01$; ****, $P < 0.0001$ by one-way ANOVA with Tukey's multiple comparisons. (D—E) Infected Caco-2 cell supernatants were collected every 24 h for 96 h. Mock-infected supernatant was collected at 96 h, but reovirus protein was not detected. Reovirus protein association with large EV, medium EV, and small EV/free virus fractions was quantified following SDS-PAGE and immunoblotting for T1L (D) or T3D (E). Error bars indicate SD. $n = 3$. *, $P < 0.05$ by one-way ANOVA with Tukey's multiple comparisons prior to normalization. Protein signal was normalized as a percentage of maximum by dividing each adjusted volume value by the highest measured value within the blot. (F) Infected-cell supernatants were harvested at 72 h, and viral infectious units associated with each EV fraction were quantified by plaque assay. Error bars indicate SD. $n = 3$. *, $P < 0.05$ by two-way ANOVA with Tukey's multiple comparisons. (G–J) Caco-2 cells were adsorbed with three individual clones of reovirus strains T1L or T3D, as indicated, at an MOI of 1 PFU/cell, and supernatants were fractionated and treated as described in Fig 4A. Infectious units were quantified by plaque assay. Plaque titer (G, I) and relative percent infectivity, quantified by dividing the treated infectious units by mock-treated infectious units and multiplying by 100 (H, J), for each sample are shown. Error bars indicate SD. "n.d." = not detected. $n$ = two titers per sample in each of three independent experiments. *, $P < 0.05$; ****, $P < 0.0001$ by two-sample unpaired T test.

fractions (**Figs 1E and 5D**). While T3D structural protein association with large EV fractions from L cells was significantly lower relative to the medium EV and small EV/free virus fractions, T3D protein association with all Caco-2-derived EV-enriched fractions was generally equivalent (**Figs 1G and 5E**). In contrast with the phenotype we previously observed in L cells, nonstructural protein σNS did not appear to associate with any EV fraction derived from Caco-2 cells infected with T1L or T3D (**S2** and **S8** **Figs**). Similar to observations with L cells, infectious reovirus was associated with all EV fractions released from Caco-2 cells (**Fig 5F**). In most cases, the distribution of infectious virus titer associated with each fraction was roughly equivalent for a given virus, though significantly lower for T1L with the medium EV fraction. Overall, we observed infectious reovirus association with EVs released from murine L cells and human Caco-2 cells, suggesting that this mechanism of EV-mediated reovirus egress is not unique to L cells and can occur in multiple cell types.

To determine whether reovirus particles are packaged inside Caco-2-derived EVs and protected from antibody neutralization, we used our plaque reduction neutralization assay (**Fig 4A**). We found that T1L was neutralized to similar levels as free reovirus when it was associated with Caco-2-derived large EVs and small EVs/free virus but was protected from neutralization when associated with medium EVs (**Fig 5G–5H**). Interestingly, T3D was efficiently neutralized to levels similar to free T3D reovirus when associated with any Caco-2-derived EV fraction (**Fig 5I–5J**). These findings suggest that T1L particles are packaged inside of medium EVs released from two cell types. However, EV-mediated protection appears to be virus-strain- and cell-type-dependent, as T3D is efficiently protected when associated with L cell-derived medium EVs, but T3D is inefficiently protected when it associates with Caco-2-derived medium EVs.

## Reovirus infection enhances EV release

To understand whether reovirus infection influences EV release on a whole-cell level, we adsorbed L cells with medium (mock) or T1L or T3D reovirus, incubated cells for 72 h, and used sequential differential centrifugation to enrich cell culture supernatants for large EV, medium EV, and small EV/free virus fractions, resuspending each in an equal volume. To compare relative amounts of protein present in each EV fraction, we used SDS-PAGE and resolved equal sample volumes (**Fig 6A**). Compared with uninfected cells, T1L- or T3D-infected cells released material containing significantly increased total protein signal in most EV fractions (**Fig 6B**). On average, we detected an approximately two-fold increase in released protein in each fraction for infected cells compared to mock-infected cells (**Fig 6C**). To select

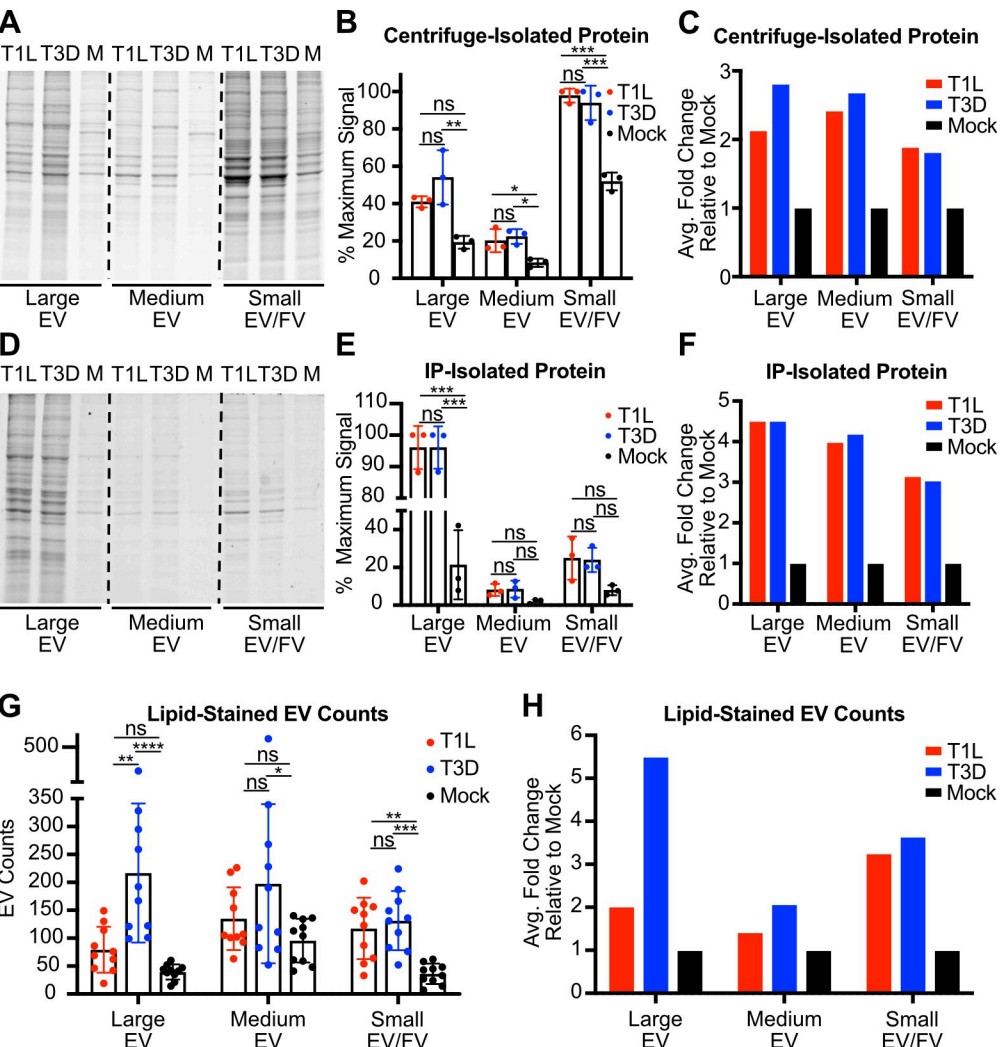

**Fig 6. Reovirus infection enhances EV release compared to uninfected cells.** L cells were adsorbed with media (mock; M) or with three individual clones of T1L or T3D reovirus at an MOI of 1 PFU/cell for 72 h. (A-C) Fractions enriched in large, medium, and small EVs were harvested from supernatants using sequential centrifugation, as previously described, then lysed. Equal lysate volumes were resolved by SDS-PAGE and Coomassie staining (A), three independent experiments were quantified (B), and they were normalized by dividing the average virus-infected value by the average mock-infected value (C). (D-F) Fractions enriched in large, medium, and small EVs were harvested from supernatants using sequential centrifugation, as previously described. Then, EVs were immunoprecipitated using annexin V nanobeads, which bind to phosphatidylserine. Equal volumes of immunoprecipitated material were resolved by SDS-PAGE and Coomassie staining (D), three independent experiments were quantified (E), and normalized by dividing the average virus-infected value by the average mock-infected value (F). (G-H) Fractions enriched in large, medium, and small EVs were harvested from supernatants using sequential centrifugation, as previously described. Each sample was resuspended in an equal volume of salt-balanced buffer, allowed to interact with a fluorescent lipid dye, loaded into the well of a Mattek dish, covered with a sterile glass cover slip, and imaged using confocal microscopy. EVs were counted in 10 random fields of view, each representing an 8 x 8 tile imaging structure (G) and normalized by dividing the average virus-infected value by the average mock-infected value (H). Error bars indicate SD. *, $P < 0.05$; **, $P < 0.01$; ***, $P < 0.001$ by one-way ANOVA with Tukey's multiple comparisons.

for EVs, we subjected large EV, medium EV, and small EV/free virus fractions to further separation using annexin V nanobead immunoprecipitation. Annexin V binds phosphatidylserine, which is present on the exterior of most EVs but is not displayed on the surface of healthy cells. After subjecting equal volumes of immunoprecipitated samples to SDS-PAGE, we found

that reovirus infection increased the protein amount associated with released large EV, medium EV, and small EV/free virus fractions relative to mock (**Fig 6D**). Although only the difference for the large EV fraction was statistically significant, on average we detected a three-fold to more than four-fold protein signal increase in EV fractions released from infected cells over the corresponding fractions from mock-infected cells (**Fig 6E–6F**). To ensure that this increase in associated protein was a result of EV release and not a general change in total protein expression triggered by reovirus infection, we adsorbed L cells with medium alone (mock) or with T1L or T3D reovirus and let infection proceed for 72 h. We then harvested the cell monolayer and quantified total protein in cell lysates by SDS-PAGE with Coommassie blue staining (**S9A Fig**). We detected no significant change nor trend towards a change in total protein expression between infected cells and uninfected cells (**S9B–S9C Fig**).

In a parallel approach to determine the effects of reovirus infection on EV release, we used a lipophilic dye, DiI, to quantify the EVs released from reovirus-infected cells compared to uninfected cells. Following adsorption with T1L, T3D, or medium (mock), we enriched large EV, medium EV, and small EV/free virus fractions. We mixed fractions with DiI to stain membranes, then imaged ten randomly selected fields of view by confocal microscopy and used the EVAnalyzer FIJI plugin to count DiI-positive puncta, which likely represent EVs (**Figs 6G and S10**) [55]. Although fields of view varied, overall lipid-stained EV counts from reovirus-infected cells were significantly higher than those from uninfected cells. The average fold change of released large EVs was notably higher from T3D-infected cells than T1L-infected cells, a five-fold change versus a two-fold change, respectively (**Fig 6H**). We hypothesize that, since T3D leads to plasma membrane disruption and apoptosis in L cells, there may be more apoptotic blebs and plasma membrane debris released from T3D-infected L cells, which are most likely to pellet in the large EV fraction. Consistent with findings from the immunoprecipitation assay, quantitation of EVs stained with lipophilic dye suggested that reovirus infection typically doubled or tripled the release of most sizes of EVs from L cells (**Fig 6F and 6H**). Collectively, these data suggest that reovirus infection enhances the release of EVs of all sizes.

## Discussion

We sought to understand mechanisms of reovirus egress and to determine whether reovirus egress strategies influence downstream infection of recipient cells. We discovered that, in addition to exiting cells as free particles, reovirus can egress from two distinct cell types in association with large EVs and medium EVs. This is true for reovirus strains that damage or disrupt cell membranes efficiently or inefficiently during infection. Nonspecific reovirus adhesion to large and medium EVs is unlikely to fully explain the association of these particles with EVs during egress. Reovirus also enhances shedding of all EV sizes from infected cells compared to uninfected cells. EV-mediated reovirus transport can ferry collective, multiparticle groups of reovirus between cells, and can initiate virus strain-dependent and cell type-specific reovirus protection from antibody-mediated neutralization (**Fig 7**). Presently, we are unable to delineate the specific EV subpopulations with which reovirus associates, though T1L and T3D have been visualized inside EVs that resemble microvesicles (**Figs 2 and S4**). Molecular and imaging approaches will reveal additional insights into the detailed mechanisms of EV-associated reovirus egress in the future.

We envision a model in which reovirus can use at least three distinct pathways to exit infected cells: (i) by membrane lysis, (ii) enclosed within medium EVs, or (iii) using a mechanism involving "sorting organelles" and "membranous carriers" [38,48]. More than one reovirus egress pathway may function in each cell, and the pathways used may vary by cell type. Bluetongue virus exits cells using both lytic and non-lytic strategies; in addition to inducing

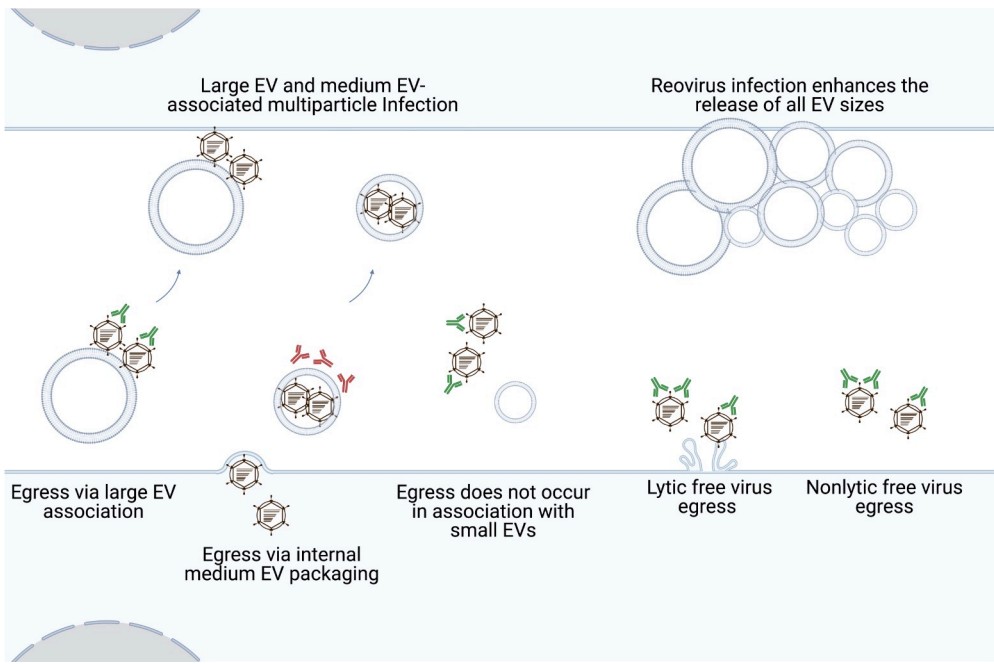

**Fig 7. Model of reovirus release and infection in association with EVs.** Our work indicates that in addition to exiting as free particles, reovirus strains that efficiently or inefficiently disrupt membranes can egress from mouse fibroblast and human colon epithelial cells in association with EVs. Reovirus particles are strain-specifically, cell type-dependently enclosed within and protected from antibody-mediated neutralization by medium EVs. Both large and medium EVs can transport multiple reovirus particles to recipient cells. Furthermore, compared to uninfected cells, reovirus infection enhances cellular release of EVs of all sizes. Created using Biorender.com.

cell lysis, bluetongue virus also buds non-lytically from the plasma membrane in multiparticle EVs that carry markers of lysosomes and exosomes [15,16]. In L cells and Caco-2 cells, T3D reovirus may employ a similar strategy in which free virus particles are released through lysis, while additional virus exits cells enclosed within medium EVs. In contrast to T3D, T1L reovirus egress occurs in the near-complete absence of membrane disruption (**Figs 1B and 5C**), which is consistent with enclosure in medium EVs but fails to explain free virus release. We propose that an additional host cell-assisted mechanism, perhaps akin to the non-lytic reovirus egress strategy described for reovirus in HBMECs, facilitates non-lytic T1L free virus egress [48]. It is also possible that free T3D virus egresses using a non-lytic mechanism. T3D does not induce significant membrane disruption at 48 h p.i.; yet we detect T3D protein association with the small EV/free virus fraction at this time point (**Figs 1B, 1G and S1**). Although T1L and T3D protein and infectious reovirus associate with large EVs derived from both L cells and Caco-2 cells (**Figs 1D–1H and 5D–5F**), we hypothesize that free reovirus particles adhere to released cell debris and large EVs after their release from cells, consistent with prior observations that large quantities of infectious virus remain associated with cellular debris following cell death induction [56]. The more efficient association of T3D with large and medium EVs derived from mock-infected cells suggests this adhesion may be more pronounced for T3D than T1L (**S3C–S3F Fig**).

Virus release in association with EVs may protect reovirus from host immune defenses and the extracellular environment. T1L and T3D reovirus released from L cells are partially shielded from antibody-mediated neutralization and proteolysis and, thus, are likely enclosed within a population of medium EVs (**Figs 4B–4E and S5**). Our findings echo those observed for BK polyomavirus, enterovirus 71, porcine reproductive and respiratory syndrome virus,

and hepatitis A virus, which can also evade antibody-mediated neutralization when they exit cells in EVs [11–13,57]. Thus, EV shielding of viruses from extracellular factors including immune defenses and proteases is a recurring strategy by which viruses from distinct families, including reovirus, may more efficiently establish or prolong infection. Given that reovirus is in clinical testing as an oncolytic therapeutic, delivery in an EV could potentially promote reovirus effectiveness as an anti-cancer agent by allowing the virus to evade neutralizing antibodies [32,58–60].

The strain-specific and cell type-dependent neutralization protection of reovirus released in association with medium EVs has potential implications for egress. While both T1L and T3D reovirus were protected in L cell-derived medium EVs, only T1L was protected in Caco-2-derived medium EVs (**Figs 4B–4E and 5G–5J**). It is possible that T3D exit from Caco-2 cells does not involve EVs; however, association of reovirus structural protein and infectious reovirus with released EV fractions following infection suggests EV involvement (**Fig 5E–5F**). T3D may be released in different EV subtypes, which are distinct from one another but similarly enriched in the medium EV fraction for L cells and Caco-2 cells. In this scenario, T3D could be packaged internally in the L cell-derived medium EV subtype and bound externally on the Caco-2-derived medium EV subtype. Precedent for a "dual EV egress" strategy exists for encephalomyocarditis virus, which exits cells in two distinct EV subtypes, one of which carries markers derived from the plasma membrane, and one of which carries markers associated with secretory autophagosomes [61]. However, given that we are currently unable to enrich solely for reovirus-containing EVs, further study is needed to discern differences between T1L-associated and T3D-associated medium EVs derived from L cells or Caco-2 cells. Another possibility, though somewhat refuted by the observation that infectious T3D reovirus associated efficiently with all EV fractions released from Caco-2 cells (**Fig 5F**), is that the efficiency with which T3D interacts with EV biogenesis pathways differs between cell types and may result in a difference in T3D's capacity to orchestrate internal or external medium EV packaging. The capacity to induce apoptosis also may influence viral escape in EVs. Reovirus induces apoptosis in L cells, a non-cancerous cell line that differs substantially from Caco-2 cells, which originate from human colorectal adenocarcinoma [40,62,63]. Due to the overexpression of anti-apoptotic molecules in the bcl-2 family and mutations in tumor suppressor genes such as p53, Caco-2 cells lack fully intact apoptosis pathways [64–68]. Sindbis virus-induced apoptosis in HeLa cells results in viral nucleocapsids and viral antigens co-localizing exclusively with EV-like structures budding from the plasma membrane of apoptotic cells [69]. Chikungunya virus-induced apoptosis leads to the formation of EV structures resembling apoptotic bodies, and when individual steps of apoptosis and EV biogenesis are inhibited, virus spread to neighboring cells is impeded [70]. Thus, differences in T3D interactions with apoptosis pathways in L cells and Caco-2 cells might influence interactions with EV biogenesis pathways and reovirus egress strategies.

The strain-specific and cell type-dependent differences in detection of reovirus nonstructural protein σNS in association with EVs may reflect reovirus biology or have potential implications for release and protection. In L cells, we observed σNS association with the small EV/free virus fraction from T1L- and T3D-infected cells and σNS association with the large and medium EV fractions from T3D-infected cells (**S2 Fig**). In Caco-2 cells, we observed a complete lack of σNS association with any EV fraction (**S8 Fig**). T3D induces apoptosis and membrane damage more efficiently than T1L (**Figs 1B, 5C, S1 and S7**). Apoptotic blebs vary broadly in size and may be detected in any EV-enriched fraction. Signal from σNS detected in large and medium EV fractions from L cells may primarily be derived from pieces of virus factories incorporated in apoptotic blebs, which would explain their absence in corresponding T1L fractions. While T3D forms globular factories, T1L forms filamentous factories that

associate with microtubules [71,72]. Factory morphology and host protein interactions also may alter σNS uptake into released EVs, particularly large and medium EVs released from the plasma membrane. In either case, if T3D particles are packaged into the same EVs as σNS, they could potentially be protected from the environment by nonstructural proteins. However, this mechanism does not explain the protection of T1L in medium EVs derived from L cells or Caco-2 cells (**Figs 4B–4C, 5G–5H, S2 and S8**). Reasons for the lack of σNS presence in T3D-infected Caco-2 cell-derived large and medium EV fractions are unclear but could correlate with differences in apoptosis pathways [64–68]. The σNS protein can bind cellular RNA [73,74]. RNA binding, protein misfolding, or ubiquitinylation and sorting by ESCRT complex proteins could serve as a mechanism to load T1L or T3D σNS into small EVs, which are known to package RNA and protein cargo [75–77]. Although exosome cargo can differ by cell type, it is unclear why σNS is present in reovirus-infected L cell-derived small EV fractions but not in any T1L- or T3D-infected Caco-2 cell-derived EV fractions (**S2 and S8 Figs**).

EVs may contribute to reovirus multiparticle infection. We observed that ~ 18% of tested infectious units released from infected cells contained multiple reovirus genotypes, with more mixed-genotype signal detected in the medium EV fraction, some in the large EV fraction, and none in the small EV/free virus fraction (**Fig 4H**). To maximize chances of forming mixed virus populations, we used an MOI at which most cells are likely to be co-infected with both WT and BC reoviruses. In previous studies using the same experimental conditions, our lab observed both WT and BC RNA co-occupying reovirus factories, sites of assembly prior to egress, and there appeared to be few limitations to WT and BC reovirus reassortment following coinfection at high multiplicity [54]. However, our HRM genotyping strategy is limited in that only infectious units containing relatively even mixtures of WT and BC RNA are likely to be detected. If an EV contains multiple particles of the same parental genome or a substantially higher ratio of one type of genome relative to the other, it is likely to be missed. We observed that 2:1, 1:1, and 1:2 mixtures of WT:BC RNA yielded melt curves that were distinct from WT and BC control melt curves, whereas 8:1, 4:1, 1:4, and 1:8 RNA ratios were more difficult to differentiate (**S6 Fig**). Thus, our technological approach may underestimate EV-mediated reovirus multiparticle infection. Additionally, coxsackievirus-containing EVs selectively package "sibling" viruses of the same parental origin, and it is possible that there is a similar unknown EV packaging bias in our system [18]. Multiparticle infection, enabled by EVs, may enhance productive virus infection. A minority of reovirus particles are thought to be infectious [78,79]. Multiparticle aggregation has been shown to increase reovirus complementation during infection, and by rescuing deleterious mutations, the overall fitness of the viral population can be enhanced [79–82]. When too few particles enter a cell, host barriers may halt virus replication [17,83–87]. In contrast with free virus, viral infectivity was enhanced when multiparticle BK polyomavirus, rotavirus, norovirus, coxsackievirus, poliovirus, and JC polyomavirus associated with EV structures [17,18,22,88,89]. For reovirus, the presence of bacteria enhances virulence, with multiparticle infection mediated through adhesion to bacteria proposed as a mechanism [81,90]. While *in vivo* studies are needed to determine the biological relevance of EV-associated infection, EV-mediated multiparticle transport may allow reovirus to complement defective genomes and overcome host cell barriers, mediating more productive infection.

On a whole-cell level, T1L and T3D infection enhances the release of all EV sizes (**Figs 6 and S10**). Large-scale viral modulation of EVs has been demonstrated for multiple viruses: poliovirus remodels intracellular membranes and lipid pools, enterovirus 71 infection induces the formation of autophagosomes, coxsackievirus proteins increase the formation of autophagosomes, Zika virus modulates exosome biogenesis proteins, and Epstein-Barr virus induces the upregulation whole-cell EV protein secretion [17,24,25,91–93]. It is currently unclear

whether reovirus upregulation of EV release is a specific response or a general cellular stress response. However, rotavirus infection upregulates EV release, and rotavirus protein co-pre-cipitation with EV biogenesis proteins *in vitro* and in human patient samples indicates intra-cellular association of rotavirus with EV biogenesis pathway molecules, which might hint at a mechanism of EV modulation [23]. By upregulating EV release, reovirus may promote its own egress. If released EVs also modulate the host immune system during reovirus infection, as they do for rotavirus, upregulation of EV release may also promote reovirus escape from immune defenses and prolong infection [23,94]. Further experimentation will reveal the mechanisms and impacts of reovirus-mediated upregulation of EV release.

Altogether, our work suggests that in addition to exiting as free independent particles, in a virus strain- and cell type-dependent manner, reovirus egresses from two distinct cell types enclosed in medium-sized, immune- and protease-protective EVs that can promote multipar-ticle infection. Although we do not yet know if this reovirus egress strategy occurs in humans or other animals, it is likely that EV-associated egress is biologically meaningful, as EV-medi-ated egress is employed by evolutionary diverse viruses. Pathogenic viruses including rotavi-rus, poliovirus, norovirus, adenovirus, coxsackievirus, and hepatitis A virus have all been shown to egress in EV-associated form. As more studies support EV-mediated egress as a via-ble route of pathogenic virus transport, greater understanding of the effects of EV-associated virus release and spread will continue to improve public health [2]. EVs can also be harnessed for therapeutic purposes, as evidenced by the rise of EV studies in the tumor microenviron-ment and the recent FDA approval employing hepatitis A virus as a tool to destroy skin cancer cells. In the case of reovirus, studies that build on the current findings may inform delivery of oncolytic reovirus to tumor sites [95].

## Materials and methods

**Cells.** Spinner-adapted L cells were grown in Joklik's minimum essential medium (JMEM; U. S. Biological) supplemented to contain 5% fetal bovine serum (FBS; Gibco), 2 mM L-glutamine (Corning), 100 U/ml penicillin (Corning), 100 mg/ml streptomycin (Corning), and 25 ng/ml amphotericin B (Corning). During infection and EV collection, L cells were cultured in serum-free JMEM. Caco-2 cells were maintained in minimum essential medium (MEM; Corn-ing) supplemented to contain 20% FBS, 100 U/ml penicillin, 100 mg/ml streptomycin, 100 U/ml non-essential amino acids (Corning), 100 U/ml HEPES buffer (Corning), 100 U/ml sodium pyruvate (Corning), and 25 ng/ml amphotericin B. During infection and EV collection, Caco-2 cells were cultured in serum-free MEM and kept in a non-polarized, non-differentiated state through maintenance splitting and seeding at sub-confluent levels. Baby hamster kidney cells expressing T7 RNA polymerase controlled by a cytomegalovirus promoter (BHK-T7) were maintained in Dulbecco's minimum essential medium (DMEM; Corning) supplemented to contain 5% FBS, 100 U/ml penicillin, 100 mg/ml streptomycin, and 1 mg/ml Geneticin, which was added every other passage. All cells were maintained at 37˚C with 5% $CO_2$.

## Viruses

Reovirus strains rsT1L (T1L), rsT3D$^I$ (T3D or WT), and rsT3D$^I$ BC (BC) were engineered using plasmid-based reverse genetics [96,97]. Strain rsT3D$^I$ is a variant of the parental rsT3D prototype strain that contains a T249I mutation that renders viral attachment protein σ1 resis-tant to proteolytic cleavage [96]. Strain rsT3D$^I$ BC is identical to rsT3D$^I$ excepting silent genetic "barcode" mutations engineered in each segment [54]. Briefly, semi-confluent mono-layers of BHK-T7 cells in 6 well plates were transfected with 10 plasmid constructs encoding T1L and T3D reovirus RNAs. After incubation at 37˚C for several days, cells were subjected to

two cycles of freezing and thawing. The resulting lysates were serially diluted and subjected to plaque assay [50]. Three individual plaques per recombinant virus strain were selected and amplified in L cells to make clonal virus stocks. Viral stock titers were quantified by plaque assay.

## Antibodies

Rabbit polyclonal reovirus antisera [98], rabbit polyclonal antisera directed against the T1L or T3D σ1 head domain [99], mouse monoclonal antibody 2H7 directed against T3D σNS and guinea pig polyclonal antisera directed against T1L σNS [98,100] were gifts from Dr. Terence Dermody. CD81 mouse monoclonal antibody (Santa Cruz Biotechnology; sc-166029) is commercially available.

## Virus replication assays

L cells ($2 \times 10^5$ cells/well) or Caco-2 cells ($4.2 \times 10^5$ cells/well) in complete medium were seeded in 12-well plates and incubated until reaching ~ 90% confluency. Cells were adsorbed in triplicate with media alone (mock) or with three clones of T1L or T3D at an MOI of 1 PFU/cell (L cells) or 5 PFU/cell (Caco-2 cells). Supernatants were aspirated and replaced with serum-free media post adsorption. Every 24 h for a total of 96 h, plates were stored at -80˚C. Then, they were freeze-thawed twice at -80˚C and room temperature. Virus titers in the resulting lysates were determined by FFA [101] and calculated based on numbers of infected cells quantified in four countable fields of view per well in duplicate wells, with countable fields containing ~ 50–500 reovirus-positive cells.

## Trypan blue membrane disruption assay

L cells ($2 \times 10^5$ cells/well) or Caco-2 cells ($4.2 \times 10^5$ cells/well) in complete media were seeded in 12-well plates and incubated until reaching ~ 90% confluency. Cells were adsorbed in triplicate with media alone (mock), with three clones of T1L or T3D at an MOI of 1 PFU/cell (L cells) or 5 PFU/cell (Caco-2 cells). Supernatants were aspirated and replaced with serum-free media post adsorption. Every 24 h for a total of 96 h, cells were gently trypsinized at 37˚C and collected via centrifugation at $100 \times g$. Cells were resuspended in equivalent volumes of PBS without $Ca^{2+}$ or $Mg^{2+}$ and 0.4% trypan blue solution (Corning), incubated for 3 min at room temperature, and then manually quantified in duplicate using a hemacytometer and a compound light microscope. Trypan-positive cells are considered to have a disrupted plasma membrane.

## Lactase dehydrogenase membrane damage assay

L cells ($2 \times 10^4$ cells/well) or Caco-2 cells ($1.9 \times 10^5$ cells/well) in complete media were seeded in 96-well black-walled plates (Greiner) and incubated until reaching ~ 90% confluency. Cells were adsorbed in triplicate with media alone (mock), with three clones of T1L or T3D at an MOI of 1 PFU/cell (L cells) or 5 PFU/cell (Caco-2 cells). Triplicate wells of uninfected cells were seeded for additional kit-specific controls, including spontaneous LDH release and maximum LDH release. Supernatants were aspirated and replaced with serum-free media post adsorption. Every 24 h for a total of 96 h, cell supernatants were harvested and plasma membrane damage was quantified in comparison with media-only negative controls and kit-provided positive controls based on the manufacturer protocol (ThermoFisher Scientific, CyQUANT LDH Cytotoxicity Assay). Absorbance at 490 and 680 was measured directly upon assay completion using the Biotek Synergy Neo 2 with accompanying Gen 5.309 software.

## Extracellular vesicle enrichment

Serum-free cell culture supernatants were collected. Cell debris was pelleted and discarded following centrifugation at $300 \times g$ for 10 min. The resulting supernatant was centrifuged at $2,000 \times g$ for 25 min to pellet large EVs, followed by centrifugation at $10,000 \times g$ for 30 min to pellet medium EVs, and then at $100,000 \times g$ for 2 h to pellet a mixed population of small EVs and free virus particles. Pelleted EV fractions were re-suspended in EV storage buffer (5M NaCl, 1M $MgCl_2$, 1M Tris pH 7.4) and stored briefly at 4°C or used immediately for assays.

## Iodixanol gradient separation of small EVs and free virus

Cell debris, large EVs, and medium EVs were cleared from supernatants by sequential differential centrifugation, as described above. The resulting supernatant was concentrated on a 2 ml 60% iodixanol cushion in 0.25 M sucrose and 10 mM Tris, pH 7.5 at $100,000 \times g$ for 4 h [102]. Following ultracentrifugation, 3 ml of iodixanol cushion plus concentrated supernatant were collected from the bottom of the ultracentrifuge tube and mixed. The resulting small EV/ free virus suspension was loaded into the bottom of a separate ultracentrifuge tube. Layers of 20%, 10%, and 5% iodixanol were added sequentially to form a gradient. The gradient was centrifuged at $100,000 \times g$ for 18 h. Then, $12 \times 1$-ml fractions were drawn starting at the top of the gradient. Resulting 1-ml fractions were washed with PBS, concentrated at $100,000 \times g$ for 2 h, gently pipetted to re-distribute the iodixanol, then concentrated again at $100,000 \times g$ for 1 h. Pelleted EV fractions were resuspended in EV storage buffer.

## Reovirus-EV immunoblotting assays

To determine reovirus protein association with large EVs, medium EVs, and small EVs, L cells ($1.5 \times 10^6$ cells/flask) or Caco-2 cells ($2.5 \times 10^6$ cells/flask) in complete media were seeded in T25 cell culture flasks and incubated until ~ 90% confluency. Cells were adsorbed in triplicate with media alone (mock), with three clones of T1L or T3D at an MOI of 1 PFU/cell (L cells) or 5 PFU/cell (Caco-2 cells), then inocula were aspirated and replaced with serum-free media. Cell culture supernatants were collected every 24 h for 96 h. At each timepoint, EV fractions were enriched via sequential differential centrifugation, as described above. Samples were resuspended in equal volumes, resolved by SDS-10% PAGE, transferred to nitrocellulose, and blocked using Pierce Protein-Free PBS Blocking buffer (ThermoScientific). Reovirus proteins were detected using polyclonal reovirus antiserum (1:1000) and LI-COR IRDye 680LT Goat anti-Rabbit (1:15,000). Small EV marker CD81 was detected using monoclonal anti-CD81 antibody (1:400) and LI-COR IRDye 800LT Goat anti-Mouse (1:15,000). T1L nonstructural protein σNS was detected using guinea pig polyclonal anti-σNS antisera (1:1,000) and LI-COR IRDye 680LT Goat anti-guinea pig (1:15,000) [100]. T3D nonstructural protein σNS was detected using mouse monoclonal anti-σNS antibody 2H7 (1:1,000) and LI-COR IRDye 800LT Goat anti-Mouse (1:15,000) [98]. Signal was detected using a Bio-Rad ChemiDoc MP Imaging System. Reovirus λ3, σNS, and CD81 protein bands were quantified with adjustment for background using the BioRad ImageLab analysis software. To compare signals from multiple experiments, the 96 h timepoint value for the small EV/free virus sample was set to 100%, and all other samples were adjusted based on this value.

## Reovirus-Mock EV association assays

L cells ($1.5 \times 10^7$ cells/flask) in complete media were seeded in T150 flasks and incubated until ~ 90% confluency. Cells were either adsorbed with T1L or T3D at an MOI of 1 PFU/cell, or with medium only (mock); for every T1L-infected or T3D-infected flask, three mock-infected

flasks were seeded and adsorbed. Inocula were aspirated and replaced with serum-free medium. After 72 h, reovirus-infected cell culture supernatants were collected and enriched in equal volumes for large EV and medium EV fractions by sequential differential centrifugation to constitute the "virus-infected EV' fraction. In parallel, mock-infected cell culture supernatants were collected and enriched in equal volumes for large EV and medium EV fractions by sequential differential centrifugation. These mock-infected EV fractions were incubated at 4°C for 2 h with an equal volume of $1 \times 10^9$ PFU/mL free reovirus particles harvested via iodixanol gradient centrifugation representing "FV input", or with EV storage buffer. Samples were then re-pelleted at respective centrifugation speeds to obtain the "FV + mock EV" and the "FV + buffer" samples. Equal sample volumes were resolved using SDS-PAGE with Coommassie blue staining, or with SDS-PAGE using polyclonal reovirus antiserum (1:1000) and LI-COR IRDye 680LT Goat anti-Rabbit (1:15,000). Signal was detected using a Bio-Rad ChemiDoc MP Imaging System. The reovirus λ3 protein band was quantified with adjustment for background using the BioRad ImageLab analysis software.

## Negative-stain transmission electron microscopy

L cells ($1.5 \times 10^7$ cells/flask) or Caco-2 cells ($2.0 \times 10^7$ cells/flask) in complete media were seeded in T150 flasks and incubated until ~ 90% confluency. Cells were adsorbed with T1L or T3D at an MOI of 1 PFU/cell, then inocula were aspirated and replaced with serum-free media. After 72 h, cell culture supernatants were collected and enriched for large EV and medium EV fractions by sequential differential centrifugation. L cell-derived fractions enriched for small EVs and free virus were separated using density-dependent gradient separation, as described. Purified samples were adhered to freshly glow discharged carbon coated 300 mesh Cu grids for 30 s followed by negative staining using 2% uranyl acetate. Transmission electron microscopy was conducted using a Tecnai T12 operating at 100 keV with an AMT nanosprint5 CMOS camera.

## Thin-section electron microscopy

L cells ($3.8 \times 10^6$ cells/dish) in complete medium were seeded in 200 mm dishes and incubated until confluent. Cells were adsorbed with medium alone or with T1L or T3D at an MOI of 1 PFU/cell. After 24 h, cells were washed thrice with pre-warmed PBS without $Ca^{2+}$ or $Mg^{2+}$ and fixed with 2.5% glutaraldehyde for 1 h at room temperature. After fixation the samples were gently lifted and embedded in 2% low-melt agar. The samples were cryoprotected in graded steps up to 30% glycerol and plunge frozen in liquid ethane. After freezing the samples were freeze-substituted at -80°C in 1.5% uranyl acetate in methanol for 48 h, then gradually raised to -30°C. Samples were infiltrated with HM-20 Lowicryl under a nitrogen atmosphere and polymerized with UV light for 48 h. Following polymerization, the samples were sectioned at a nominal thickness of 70 nm on a Leica UC7 ultramicrotome and imaged as described above.

## Quantitation of released extracellular vesicles

L cells ($1.5 \times 10^7$ cells/flask) in complete medium were seeded in T150 flasks and incubated until ~ 90% confluency. Cells were adsorbed with medium alone (mock) or with three clones of T1L or T3D at an MOI of 1 PFU/cell, then inocula were aspirated and replaced with serum-free medium. After 72 h, cell culture supernatants were collected and enriched for large EV, medium EV, and small EV/free virus fractions by sequential differential centrifugation. A subset of samples was resuspended in equal volumes of EV storage buffer, resolved by SDS-10% PAGE, and stained with PageBlue Protein Staining Solution (Thermo). Gels were imaged using a Bio-Rad ChemiDoc MP Imaging System, and proteins in entire lanes were quantified

using Bio-Rad ImageLab analysis software. To compare signals from multiple experiments, protein signal was normalized as a percentage of maximum by dividing each adjusted volume value by the highest measured value within the blot.

A subset of samples, which were subjected to annexin V immunoprecipitation (Miltenyi Biotec Annexin V Microbead Immunoprecipitation kit), was resuspended in Annexin V Binding Buffer and incubated with Annexin V Microbeads for 2 h at 4˚C with rotation. The samples were applied to a Miltenyi Biotec MS Column on a Miltenyi Biotec MiniMACS Separator and eluted in equal volumes of EV storage buffer. Eluates were resolved by SDS-10% PAGE, stained with PageBlue Protein Staining Solution, and quantified as described above.

A subset of samples, which were analyzed using confocal microscopy, was resuspended in equal volumes of EV storage buffer and stored on ice. Samples were individually mixed with an equal volume of 1 µg/ml of 1,1'-Dioctadecyl-3,3,3',3'-Tetramethylindocarbocyanine Perchlorate (DiI; Invitrogen), incubated for 15 min on ice, then loaded into the well of a Mattek 35 mm dish with a 1.4 mm glass coverslip. Each dish was incubated for 5 min at room temperature and then imaged on a Zeiss LSM 880 microscope under a 63X oil lens. A total of 10 fields of view per sample, each comprising an 8 x 8 stitched tile, were imaged. DiI-positive puncta in each field were counted using the EVAnalyzer Fiji Software plugin [55], with threshold settings applied uniformly across separate EV fractions to T1L-infected, T3D-infected, and mock-infected samples.

## EV neutralization protection assays

L cells (1.5 x $10^7$ cells/flask) or Caco-2 cells (2.0 x $10^7$ cells/flask) in complete media were seeded in T150 flasks and incubated until ~ 90% confluency. Cells were adsorbed with media alone (mock) or with three clones of T1L or T3D at an MOI of 1 PFU/cell (L cells) or 5 PFU/cell (Caco-2 cells), then inocula were aspirated and replaced with serum-free media. After 72 h, cell culture supernatants were enriched for large EVs, medium EVs, and small EVs/free virus by sequential differential centrifugation. Samples were resuspended in EV storage buffer and divided. One half of the sample was mock-treated with serum-free medium, and the other half of the sample was treated with T1L or T3D σ1 head-specific antisera (1:100) for 2 h at 4˚C. Then, virus titer in each sample was quantified by plaque assay using the dilution yielding a range of 40–70 plaques to calculate the titer. The percent infectivity level retained post-neutralization was determined by dividing the treated sample titer by the titer of its untreated counterpart.

## EV protease protection assays

L cells (1.5 x $10^7$ cells/flask) in complete media were seeded in T150 flasks and incubated until ~ 90% confluency. Cells were adsorbed with medium alone (mock) or with three clones of T1L or T3D at an MOI of 1 PFU/cell, then inocula were aspirated and replaced with serum-free medium. After 72 h, cell culture supernatants were enriched for large EVs, medium EVs, and small EVs/free virus by sequential differential centrifugation. Samples were resuspended in EV storage buffer and divided. One half of the sample was mock treated with serum-free medium, and the other half of the sample was treated with 20 µg/mL of chymotrypsin (Sigma Aldrich) for 1 h at 37˚C. After incubation, chymotrypsin activity was neutralized using 2% total volume of 100 mM PMSF. Samples were resuspended in equal volumes, resolved by SDS-10% PAGE, transferred to nitrocellulose, and blocked using Pierce Protein-Free PBS Blocking buffer (ThermoScientific). The conversion of EV-associated reovirus from virion to ISVP, evidenced by the loss of σ3 and µ1C proteins, was quantified using SDS-PAGE with immunoblotting using polyclonal anti-reovirus serum (1:1000). Signal was detected using a Bio-Rad ChemiDoc

MP Imaging System. Reovirus σ3 and μ1C protein bands were quantified with adjustment for background using the BioRad ImageLab analysis software. The percent of σ3 and μ1C protein retained post protease treatment was determined by dividing the treated sample protein signal by the protein signal of its untreated counterpart.

## High-resolution melt analysis of genotype mixing

L cells ($4 \times 10^5$ cells/well) in complete medium were seeded in 6-well plates and incubated until ~ 90% confluency. Cells were adsorbed with medium alone (mock) or co-infected with three independent dilutions of WT and BC reovirus at an MOI of 10 PFU/cell, then inocula were aspirated and replaced with serum-free medium. After 24 h, cell culture supernatants were collected and enriched for large EV, medium EV, and small EV/free virus fractions by sequential differential centrifugation. Infectious units were then isolated by plaque assay. A total of 24 well-separated plaques per fraction per replicate were picked and amplified in L cell monolayers in 24-well plates for 2 days. RNA was extracted using TRIzol (Invitrogen), reverse transcribed using random hexamers, and genotyped using HRM, as previously described, using primers specific for the L2 segment [54]. Each sample genotype was called by Applied Biosystems High Resolution Melt Software v3.2 and visually verified by comparison with control reactions containing WT RNA, BC RNA, and mixtures (1:2, 1:1, and 2:1) of WT and BC RNA.

## Statistical analyses

GraphPad Prism version 10 was used for all statistical analyses. The statistical analyses used are indicated in each figure legend and are denoted separately for each data set. Statistical tests were chosen in consultation with a biostatistician.

## Supporting information

**S1 Fig. Reovirus plasma membrane disruption is strain specific in L929 cells.** L cells were adsorbed with medium (mock) or with three individual clones of T1L or T3D reovirus at an MOI of 1 PFU/cell. Cell membrane disruption was quantified for T1L-, T3D-, and mock-infected cells every 24 h for 96 h using an LDH assay. A medium-only negative control and a kit-specific positive control quantified in triplicate at 96 h are shown. Error bars indicate SD. $n = 3$. **, $P < 0.01$; ****, $P < 0.0001$ by one-way ANOVA with Tukey's multiple comparisons. (TIF)

**S2 Fig. Reovirus nonstructural proteins associate with EV fractions in a strain-specific manner.** (A-C) L cells were adsorbed with three individual clones (C1-C3) of T1L or T3D reovirus at an MOI of 1 PFU/cell for 72 h. Reovirus nonstructural protein association with large EV, medium EV, and small EV/free virus fractions was quantified following SDS-PAGE and immunoblotting (A) for T1L σNS (B) or T3D σNS (C). Error bars indicate SD. $n = 3$. *, $P < 0.05$; ***, $P < 0.001$; ****, $P < 0.0001$ by one-way ANOVA with Tukey's multiple comparisons prior to normalization. Protein signal was normalized as a percentage of maximum by dividing each adjusted volume value by the highest measured value for each clone within the blot. (TIF)

**S3 Fig. Little reovirus spontaneously associates with EVs.** (A-F) L cells were adsorbed with three individual clones of T1L or T3D reovirus at an MOI of 1 PFU/cell. In parallel, triple the amount of L cells were adsorbed with medium (mock). After 72 h, large and medium EVs were harvested via centrifugation from reovirus-infected cells to constitute the "virus-infected

EV" samples and from mock-infected cells. 1 x $10^9$ total PFU of free reovirus particles were mixed and incubated with large or medium EVs from mock-infected cells (mock EVs) or with EV storage buffer (buffer), then re-pelleted at respective centrifugation speeds. Equal volumes of all T1L (A, C, D) and T3D (B, E, F) samples were resolved by SDS-PAGE and Coomassie staining (A-B) or by SDS-PAGE with immunoblotting using anti-reovirus serum (C-F). The spontaneous association of free reovirus with mock large and medium EVs was quantified and compared to free T1L virus input (D) or free T3D virus input (F). Error bars indicate SD. $n = 3$. ****, $P < 0.0001$ by two-sample unpaired T test.
(TIF)

**S4 Fig. EV-mediated reovirus egress is consistent with microvesicle biogenesis.** Transmission electron microscopy of T1L-infected (A) or T3D-infected (B) L cells at 24 h p.i. Arrows point to viral particles observed near bleb-like structures budding from the plasma membrane in or around cells. Scale bar = 200 nm.
(TIF)

**S5 Fig. EVs partially protect reovirus from protease treatment.** (A-D) L cells were adsorbed with three individual clones of T1L or T3D reovirus at an MOI of 1 PFU/cell for 72 h. Large, medium, and small EV/free virus fractions were harvested via centrifugation and each split into two aliquots containing equal volumes. One aliquot was left untreated (-), and the other aliquot was treated with 20 μg/mL of chymotrypsin (+). Reovirus T1L (A-B) and T3D (C-D) σ3 and μ1C proteins were visualized by and quantified by SDS-PAGE and immunoblotting using anti-reovirus serum. Shown are representative immunoblots for each virus strain alongside values quantified for the three clones. Error bars indicate SD. $n = 3$. *, $P < 0.05$; **, $P < 0.01$ by one-way ANOVA with Tukey's multiple comparisons.
(TIF)

**S6 Fig. Sensitivity of high-resolution melt analysis.** Normalized melt curves for control RNA from WT (red), BC (blue), and mixtures of WT and BC (green) at the indicated ratios are shown.
(TIF)

**S7 Fig. Reovirus plasma membrane disruption is strain-specific in Caco-2 cells.** Caco-2 cells were adsorbed with medium (mock) or with three individual clones of T1L or T3D reovirus at an MOI of 5 PFU/cell. Cell membrane disruption was quantified for T1L-, T3D-, and mock-infected cells every 24 h for 96 h using an LDH assay. A medium-only negative control and a kit-specific positive control quantified in triplicate at 96 h are shown. Error bars indicate SD. $n = 3$. ***, $P < 0.001$; ****, $P < 0.0001$ by one-way ANOVA with Tukey's multiple comparisons.
(TIF)

**S8 Fig. Reovirus nonstructural protein association with EV fractions is cell type dependent.** Caco-2 cells were adsorbed with three individual clones (C1-C3) of T1L or T3D reovirus at an MOI of 5 PFU/cell for 72 h. Reovirus nonstructural protein association with large EV, medium EV, and small EV/free virus fractions was quantified following SDS-PAGE and immunoblotting for T1L σNS or T3D σNS.
(TIF)

**S9 Fig. Reovirus infection does not significantly alter whole cell protein expression.** L cells were adsorbed with medium (mock) or with three individual clones of T1L or T3D reovirus at an MOI of 1 PFU/cell for 72 h. (A-C) Cells were lysed in RIPA buffer, and lysates were resolved by SDS-PAGE and Coomassie staining (A), three independent experiments were quantified

(B), and normalized by dividing the average virus-infected value by the average mock-infected value (C). Error bars indicate SD, $n$ = 3. Comparisons by one-way ANOVA with Tukey's multiple comparisons.
(TIF)

**S10 Fig. Reovirus infection enhances EV release.** Representative confocal images described in **Fig 6G–6H** are displayed for a single field of view, which is made up of an 8 x 8 tile imaging structure under 63X oil immersion.
(TIF)

## Acknowledgments

We thank Jenny Schafer at the Vanderbilt University Cell Imaging Shared Resource Core for assistance with sample preparation, imaging, and analysis. We thank the Vanderbilt High Throughput Screening Facility and the VUMC Cell and Developmental Biology Resource Core for training and access to equipment. We thank the laboratories of Dr. Terence Dermody for reovirus antibodies, antisera, and consultation on the LDH assay, Dr. Alissa Weaver for consultation on iodixanol gradient separation, and Dr. Geoffrey Holm for consultation on annexin V immunoprecipitation. We thank Dr. James C. Slaughter for biostatistics expertise. We thank Dr. Terence Dermody, Julia Diller, Alejandra Flores, Dr. Danica Sutherland, Dr. Gwen Taylor, and Dr. Timothy Thoner for critical review of the manuscript.

## Author Contributions

**Conceptualization:** Sydni Caet Smith, Kristen M. Ogden.

**Formal analysis:** Sydni Caet Smith.

**Funding acquisition:** Kristen M. Ogden.

**Investigation:** Sydni Caet Smith, Evan Krystofiak.

**Methodology:** Sydni Caet Smith.

**Supervision:** Kristen M. Ogden.

**Validation:** Sydni Caet Smith.

**Visualization:** Sydni Caet Smith.

**Writing – original draft:** Sydni Caet Smith.

**Writing – review & editing:** Evan Krystofiak, Kristen M. Ogden.

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
