## [Decision Letter · Decision Letter 0]

6 Oct 2023

Dear Dr. Ogden,

Thank you very much for submitting your manuscript "Mammalian orthoreovirus can exit cells in extracellular vesicles" for consideration at PLOS Pathogens. As with all papers reviewed by the journal, your manuscript was reviewed by members of the editorial board and by several independent reviewers. In light of the reviews (below this email), we would like to invite the resubmission of a significantly-revised version that takes into account the reviewers' comments.

Thank you very much for submitting the manuscript "Mammalian orthoreovirus can exit cells in extracellular vesicles" (PPATHOGENS-D-23-01432) for consideration by PLOS Pathogens. The manuscript was reviewed by three individuals with considerable expertise in reovirus biology and/or the potential role of EVs in virus replication and spread. Overall, all three were highly complementary in their assessment of the work and the potential significance of its findings. However, all the reviewers raised major issues that they felt should be addressed before the manuscript is further considered for publication. To highlight major issues, Reviewer 1 was concerned whether trypan blue exclusion was a sufficiently sensitive assay method to evaluate membrane disruption and also desired information on the presence of nonstructural proteins in the EV populations. Reviewer 2 was concerned about the stability of EVs during EM analysis and whether this gave raise to incorrect estimates in the number of free versus EV associated reovirus particles produced by infected cells. Reviewer 2 also questioned data analysis that led to the conclusion that T1L and T3D infected cells differed quantitatively in their generation of virus-associated EVs. In addition to other concerns, Reviewer 3 wanted data addressing whether free virus could associate with isolated EVs, an important issue in ruling out artifactual measurements in the proportion of free virus versus EV associated virus. Given the issues raised by the reviewers, responses will likely require the addition of further data and extensive modification of the manuscript. If you choose to submit a revised version (which is encouraged), please include a point-by-point response to the reviewers comments and concerns addressing how the manuscript was modified (or why not). Thank you again for submitting your manuscript to the journal. We look forward to receiving a revised manuscript in the near future.

We cannot make any decision about publication until we have seen the revised manuscript and your response to the reviewers' comments. Your revised manuscript is also likely to be sent to reviewers for further evaluation.

Sincerely,

John T. Patton, PhD

Academic Editor

PLOS Pathogens

Guangxiang Luo

Section Editor

PLOS Pathogens

Kasturi Haldar

Editor-in-Chief

PLOS Pathogens

orcid.org/0000-0001-5065-158X

Michael Malim

Editor-in-Chief

PLOS Pathogens

orcid.org/0000-0002-7699-2064

Thank you very much for submitting the manuscript "Mammalian orthoreovirus can exit cells in extracellular vesicles" (PPATHOGENS-D-23-01432) for consideration by PLOS Pathogens. The manuscript was reviewed by three individuals with considerable expertise in reovirus biology and/or the potential role of EVs in virus replication and spread. Overall, all three were highly complementary in their assessment of the work and the potential significance of its findings. However, all the reviewers raised major issues that they felt should be addressed before the manuscript is further considered for publication. To highlight major issues, Reviewer 1 was concerned whether trypan blue exclusion was a sufficiently sensitive assay method to evaluate membrane disruption and also desired information on the presence of nonstructural proteins in the EV populations. Reviewer 2 was concerned about the stability of EVs during EM analysis and whether this gave raise to incorrect estimates in the number of free versus EV associated reovirus particles produced by infected cells. Reviewer 2 also questioned data analysis that led to the conclusion that T1L and T3D infected cells differed quantitatively in their generation of virus-associated EVs. In addition to other concerns, Reviewer 3 wanted data addressing whether free virus could associate with isolated EVs, an important issue in ruling out artifactual measurements in the proportion of free virus versus EV associated virus. Given the issues raised by the reviewers, responses will likely require the addition of further data and extensive modification of the manuscript. If you choose to submit a revised version (which is encouraged), please include a point-by-point response to the reviewers comments and concerns addressing how the manuscript was modified (or why not). Thank you again for submitting your manuscript to the journal. We look forward to receiving a revised manuscript in the near future.

Reviewer's Responses to Questions

**Part I - Summary**

Reviewer #1: The manuscript from Smith, Krystofiak and Ogden provides data supporting the hypothesis that mammalian reovirus exits from cells in extracellular vesicles. The authors provide data showing that reoviruses associate with small medium, and large EVs, but that only medium EVs provide protection from neutralization by antibodies, suggesting that the virions are protected within medium EVs, but not within small or large EVs. Further experiments show that small EV fractions contain free virus. The authors further describe that EV protection from neutralization is both strain and cell-type specific. This is a well-written, easy to follow paper. The experiments and data supporting the authors conclusions are for the most part robust. The findings will be of interest to the field and more generally within the virology community.

Major strengths include: (i) the experiments are rigorous and well-controlled. (ii) The data suggest that reovirus virions are released within medium EVs and that multiple particles can be contained with a single EV leading to more efficient infection. (iii) The phenotype of EV release is described in more than one cell type (L-cells and Caco-2 cells).

Weaknesses are:

(i) Ultimately it remains unclear if reovirus particles are released within intact medium EVs and are thus protected from neutralizing antibodies OR if they are perhaps protected by some other mechanism(s). (ii) Although the experiments were carried out in two different cell types, they are both transformed cells and may not represent what would happen in vivo. (iii) No experiments were carried out to show that virions associated with EVs were more infectious than free virions. Experiments similar to those described by Chen et al. 2015. Cell 160:619–630 to compare the infectivity within EVs versus free virus would be strengthen the paper.

Reviewer #2: Smith and colleagues present an elegant study investigating the role of extracellular vesicles in orthoreovirus egress. Mammalian orthoreovirus is a non-enveloped dsRNA virus with a segmented genome. Mechanisms of its egress are poorly understood. Much of the earlier literature has assumed that these viruses egress lytically, releasing from cells as free viral particles. A recent paper in 2020 provided evidence for these viruses egressing as free particles through a non-lytic mechanism: lysosomal exocytosis

Here in this manuscript, the authors investigate the egress of two strains of mammalian orthoreovirus, T3D and T1L. These two strains apparently differ in their pathogenesism tropism and ability to induce apoptosis. According to the authors citation of previous literature, T3D induces apoptosis more efficiently than T1L.

Here the authors carry out a combination of experiments measuring plasma membrane permeability ( with trypan), harvesting of different sizes of extracellular vesicles and quantification of associated virions, virion neutralization assays along with nice electron microscopy data. They conclude that both strains can be released in extracellular vesicles, protecting these virions from neutralization. Also they demonstrate that infection with these strains stimulates overall EV secretion from cells.

Overall this is a nice study that will be highly significant to the dsRNA virus community by highlighting the novel role of extracellular vesicles in the reovirus lifecycle.

A few suggestions for the authors that need clarification:

1. Figure 1B versus Figure 1D and 1F: It seems that for both strains a significant majority of the intracellular viral pools egresses by EVs at 48 hours ( and perhaps a bit earlier). By 48 hrs, the blots show almost equivalent amounts of virus of both strains has been released via medium EVs and small EVs/free virus ( see 1D and 1F). If anything T3D- the supposedly more lytic virus, has released medium EVs with virus quite early. . The difference in cell permeability at 48hrs between T3D and T1L is miniscule ( regardless of what the authors suggest with a significance asterisk)-

By 72 hours, the amount of lysis in T3D population is ~50%, but there is no INCREASE in the level of T3D being released in small EV/free virus from 0-48 hrs, versus 0-72 hrs.

Please explain.

2. How do the authors know that their vesicles did not lyse when they did the Ems? Likely that those viruses on the outside of the vesicles were once inside and during the EM procedure , the vesicle shrunk/broke and the viruses were released.

Please indicate this possibility in the text if it cannot be ruled out. And indeed if it cannot be ruled out, then the authors data in Figures 3 and 4 and of course the cartoon need to be totally revised.

Reviewer #3: In this paper, Smith et al explore mechanisms by which mammalian reovirus exists host cells. While a specialized non-lytic exit pathway has been shown to occur in endothelial cells, whether this mechanism is universal is not understood. The authors report using two different reovirus strains that release of each of these strains from cultured fibroblasts and intestinal epithelial cells occurs independently of cell lysis. Instead, at least some proportion of the virus seems to exit cells in association with extracellular vesicles of various kinds. The remainder of the virus is released as free virus. When virus is associated with vesicles, virus seems resistant to neutralization. However, this neutralization efficiency varies with cell type from which the vesicles are derived and strain of infection. The authors demonstrate that multiple viruses can be delivered when associated with vesicles and therefore could launch an infection that allows for the possibility for reassortment.

The data presented in the manuscript are clear and support the authors’s conclusion that vesicle associated release of virus is one mechanism of reovirus release. The paper is also well written. It remains unclear how virus associates with extracellular vesicles and why there exist cell type and virus strain specific differences in the nature of association with extracellular vesicles. The authors could better characterize whether the virus is inside or outside the vesicles (or what proportion is where), whether vesicle association is an active process or an artefact of purification or simply a passive effect of virus-membrane association. Finally, it appears that even when the particles are vesicle associated, at least half the virus is released also as free virus. Thus, the relevance of this mode of release and transmission between cells remains unknown.

**Part II – Major Issues: Key Experiments Required for Acceptance**

Reviewer #1: 1) The authors use trypan-blue exclusion as a means to determine membrane disruption. The disruption of cell membranes by viruses is likely not all-or-none, but rather is a continuum – the reovirus μ1 protein is a potential viroporin and the relative amounts of free μ1 may determine the degree of membrane permeability. The trypan-blue exclusion assay is less sensitive than similar assays that use fluorescent dye uptake (perhaps combined with flow cytometry) and is necessarily subjective using the methods described. Trypan-blue can also lead to direct effects on dead cells that lead to them rupturing and having a dim uptake of dye. The concern would be that potentially membrane disruption is occurring in the T1L-infected cells, but the level is below the threshold of sensitivity for the Trypan-blue assay. This may be important as membrane disruption does occur in living cells and may be a driver of EV generation – budding or blebbing of membranes that contain wounds may generate EVs that are not intact. It would be interesting to compare the results from Trypan-blue with perhaps propidium iodide staining.

2) The authors show nicely that viral structural proteins associate with large, medium, and small EVs. However, it would be interesting to see if nonstructural proteins are also present, e.g., μNS or σNS. This may be pertinent, as it is feasible that virions buried within remnants of viral factories may be protected from neutralizing antibodies, despite the fact they are not contained inside an EV. Excluding the presence of nonstructural proteins from EVs would remove this concern.

Reviewer #2: 1. Figure 1B versus Figure 1D and 1F: It seems that for both strains a significant majority of the intracellular viral pools egresses by EVs at 48 hours ( and perhaps a bit earlier). By 48 hrs, the blots show almost equivalent amounts of virus of both strains has been released via medium EVs and small EVs/free virus ( see 1D and 1F). If anything T3D- the supposedly more lytic virus, has released medium EVs with virus quite early. . The difference in cell permeability at 48hrs between T3D and T1L is miniscule ( regardless of what the authors suggest with a significance asterisk)-

By 72 hours, the amount of lysis in T3D population is ~50%, but there is no INCREASE in the level of T3D being released in small EV/free virus from 0-48 hrs, versus 0-72 hrs.

Please explain.

2. How do the authors know that their vesicles did not lyse when they did the Ems? Likely that those viruses on the outside of the vesicles were once inside and during the EM procedure , the vesicle shrunk/broke and the viruses were released.

Please indicate this possibility in the text if it cannot be ruled out. And indeed if it cannot be ruled out, then the authors data in Figures 3 and 4 and of course the cartoon need to be totally revised.

Reviewer #3: 1. Since extracellular vesicles can be produced from non-infected cells, what is the effect of mixing purified virus with extracellular vesicles? Can the authors demonstrate that virus will not associate with membranes under these conditions? This speaks to the specificity and mechanism of membrane interactions

2. Can the authors use techniques other than neutralization to demonstrate that virus is present inside of the vesicles? Perhaps accessibility to a protease? If the virus is not neutralized by �1 specific sera, will it infect cells that lack �1 receptors? In other words, how does vesicle associated virus launch infection? What other antisera against reovirus capsids fail to neutralize virus? In conditions where vesicle associated virus was not protected from neutralization, were other antibody concentrations used? Can neutralization resistant virus become sensitive upon solubilization/disruption of the membrane?

3. Does the spread of virus in a population where infection is initiated at low MOI dependent on EV-associated virus? What is the consequence of blocking extracellular formation. Are inhibitors of this process effective at reducing virus release or cell-to-cell spread, or coinfection efficiency? Addressing this type of question will address the relevance of this exit mechanism.

**Part III – Minor Issues: Editorial and Data Presentation Modifications**

Reviewer #1: The particles shown in Figure 2 in the Large and medium EVs differ in morphology – it looks like there are some empty particles and some full particles and other particles differ again. It would be helpful to indicate in the Figure legend what the arrowheads indicate, but also it may be interesting to classify the different types of particles that are present in the large versus medium EVs.

Line 135, the authors state that the initial cell binding varied between the two strains. Please provide the data or a reference

Reviewer #2: Paper can benefit from editing and streamlining the figures.

Reviewer #3: N/A

PLOS authors have the option to publish the peer review history of their article (what does this mean?). If published, this will include your full peer review and any attached files.

Reviewer #1: No

Reviewer #2: No

Reviewer #3: No
---

## [Decision Letter · Decision Letter 1]

2 Jan 2024

Dear Dr. Ogden,

We are pleased to inform you that your manuscript 'Mammalian orthoreovirus can exit cells in extracellular vesicles' has been provisionally accepted for publication in PLOS Pathogens.

Best regards,

Guangxiang George Luo, M.D./MPH

Section Editor

PLOS Pathogens

Kasturi Haldar

Editor-in-Chief

PLOS Pathogens

orcid.org/0000-0001-5065-158X

Michael Malim

Editor-in-Chief

PLOS Pathogens

orcid.org/0000-0002-7699-2064

Reviewer Comments (if any, and for reference):

Reviewer's Responses to Questions

**Part I - Summary**

Reviewer #1: The authors have done a nice job in addressing the critiques I included in my first review. They used a fluorescent LDH assay to confirm the Trypan Blue exclusion assay, and they included assays to address the possibility that nonstructural proteins may be incorporated with particles inside EVs. I understand that addressing the infectivity issue may take additional experimentation beyond the scope of this paper, which is already quite lengthy. This is not a critique, but a comment: LDH release requires large pores or breaches of the membrane (its's a 140 KDa tetramer), whereas Trypan blue is only 960 Da. Some of the newer fluorescent dyes for detecting membrane breaches are smaller still, e.g., DRAQ7 is in the 400-500 Da size range. Small pores in membranes may drive EV formation - might be worth investigating using smaller fluorescent dsDNA binding dyes or looking for Calcium fluxes.

Overall, this is an important paper that advances our knowledge of mammalian reovirus egress pathways. The data are convincing and the paper is well written.

Reviewer #2: Authors have responded thoughtfully to my previous comments and have provided an elegant study that is suitable for publication.

Reviewer #3: In this paper, Smith et al describe a new mechanism by which mammalian reovirus exists host cells. The authors demonstrate that release occurs in extracellular vesicles and is independent of cell lysis. These are new and important findings and the experiments are well designed and presented.

In the previous submission, I made 3 major comments that the authors should address. The authors have satisfactorily addressed 2 of my 3 comments. I am satisfied by this response. I am also convinced by their argument that the 3rd comment is not easy to address.

**Part II – Major Issues: Key Experiments Required for Acceptance**

Reviewer #1: Not applicable.

Reviewer #2: None

Reviewer #3: N/A

**Part III – Minor Issues: Editorial and Data Presentation Modifications**

Reviewer #1: (No Response)

Reviewer #2: None

Reviewer #3: N/A

PLOS authors have the option to publish the peer review history of their article (what does this mean?). If published, this will include your full peer review and any attached files.

Reviewer #1: No

Reviewer #2: No

Reviewer #3: No

---

## [Editor Report · Acceptance letter]

6 Jan 2024

Dear Dr. Ogden,

We are delighted to inform you that your manuscript, "Mammalian orthoreovirus can exit cells in extracellular vesicles," has been formally accepted for publication in PLOS Pathogens.

Best regards,

Michael Malim

Editor-in-Chief

PLOS Pathogens

orcid.org/0000-0002-7699-2064